# Differential cortical layer engagement during seizure initiation and spread in humans

Pierre Bourdillon [1,2,3] ✉, Liankun Ren [4,5], Mila Halgren[6],
Angelique C. Paulk [1], Pariya Salami[1], István Ulbert [7,8,9], Dániel Fabó [9],
Jean-Rémi King [10], Kane M. Sjoberg [1,11], Emad N. Eskandar[12],
Joseph R. Madsen[13], Eric Halgren [14] & Sydney S. Cash [1]

Despite decades of research, we still do not understand how spontaneous human seizures start and spread – especially at the level of neuronal microcircuits. In this study, we used laminar arrays of micro-electrodes to simultaneously record the local field potentials and multi-unit neural activities across the six layers of the neocortex during focal seizures in humans. We found that, within the ictal onset zone, the discharges generated during a seizure consisted of current sinks and sources only within the infra-granular and granular layers. Outside of the seizure onset zone, ictal discharges reflected current flow in the supra-granular layers. Interestingly, these patterns of current flow evolved during the course of the seizure – especially outside the seizure onset zone where superficial sinks and sources extended into the deeper layers. Based on these observations, a framework describing cortical-cortical dynamics of seizures is proposed with implications for seizure localization, surgical targeting, and neuromodulation techniques to block the generation and propagation of seizures.

Epilepsy affects 1% of the population and has a major impact on the quality of life of patients[1,2]. In almost one in three cases, epilepsy is drug-resistant and, if focal, can be treated by surgery by removing or destroying the region of the brain responsible for seizure generation, a brain area also called the ictal or seizure onset zone[2,3]. The most effective method for identifying this area remains the neural recording of seizures, often by intracranial electrodes implanted for several days[4,5]. These electrodes measure the local field potentials of large

populations of neurons, enabling the identification of pathological networks involved in the propagation and continuation of seizures[4,6–9]. Nevertheless, our understanding of the cellular and circuit elements involved in seizure generation and spread remains rudimentary. Technological developments have recently led to the use of new generations of electrodes that record the activity of single neurons[4,10], or small populations of neurons at spatial scales of less than 1 mm[11–13]. These newer electrodes include laminar electrodes which penetrate

[1]Department of Neurology, Center for Neurotechnology and Neurorecovery, Massachusetts General Hospital and Harvard Medical School, Boston, MA, USA. [2]Department of Neurosurgery, Hospital Foundation Adolphe de Rothschild, Paris, France. [3]Integrative Neuroscience and Cognition Center, Paris Cité University, Paris, France. [4]Department of Neurology, Xuanwu Hospital, National Center for Neurological Disorders, Clinical Center for Epilepsy, Capital Medical University, Beijing, China. [5]Chinese Institute for Brain Research, Beijing, China. [6]Brain and Cognitive Sciences Department and McGovern Institute for Brain Research, Massachusetts Institute of Technology, Cambridge, MA, USA. [7]HUN-REN, Research Center for Natural Sciences, Institute of Cognitive Neuroscience and Psychology, Budapest, Hungary. [8]Faculty of Information Technology and Bionics, Péter Pázmány Catholic University, Budapest, Hungary. [9]Department of Neurosurgery and Neurointervention, Faculty of Medicine, Semmelweis University, Budapest, Hungary. [10]Laboratoire des Systèmes Perceptifs, Département d'études cognitives, École normale supérieure, PSL University, CNRS, Paris, France. [11]Harvard College, Cambridge, MA 02138, USA. [12]Department of Neurological Surgery, Albert Einstein College of Medicine – Montefiore Medical Center, Bronx, NY, USA. [13]Department of Neurosurgery, Boston Children Hospital, Harvard Medical School, Boston, MA, USA. [14]Departments of Radiology and, Neurosciences, University of California, San Diego, San Diego, CA, USA. ✉e-mail: pierre.bourdillon@neurochirurgie.fr

the pia to record perpendicular to the cortical surface across the different layers of the grey matter[14,15]. These laminar electrodes allow the recording of all six cortical sheets through 24 recording channels and can be used over several weeks, offering the possibility of a layer-by-layer study of cortical microcircuitry during the generation of seizures and their propagation. This is of particular interest as prior studies point to the possibility that seizure generation and propagation are localized to different cortical layers[15–18]. To study seizure propagation across cortical layers, injection of antagonists of γ-aminobutyric acid A (picrotoxin) induced seizures chemically during in vitro studies in non-human animal cortex. This approach showed that the inhibition defect caused by the chemical intervention led to the generation of horizontally propagating seizures with a crucial role of layer 5 in that process[19]. In vivo, studies using bi-photonic imaging have also shown the role of different cortical layers in the generation and propagation of seizures. In 4-aminopyridine–induced neocortical discharges and seizures in mice, it was shown that seizure initiation by this method appeared to take place in the supra-granular layers (superficial layers)[20]. In contrast, another study in mice using two-photon calcium imaging with local field potential recordings to map cortical layers at a cellular level, found different results that suggest the horizontal spread of locally induced seizures (by 4-aminopyridine or picrotoxin) relies on a supra-granular invasion followed by deep-layer recruitment[21]. Furthermore, several animal studies have been conducted using laminar electrodes to record both local field potentials and multi-unit activity to identify the role of different cortical layers in seizure generation. In a set of animal studies using flurothyl to induce seizures, all six cortical layers were implicated in seizure generation[22]. On the other hand, in neonatal mice, seizures induced by 4-aminopyridine resulted in the generation of seizures in the infra-granular layers 5 and 6 while seizure propagation is observed within the supra-granular layers 2 and 3, mainly due to the activation of interneurons. Pyramidal cells were only involved in the infragranular layers during seizure generation[23,24]. In humans, one of the very few laminar studies of epileptic activity in cortex was carried out ex-vivo, demonstrating a role of supra-granular layers in the generation of seizure activity[25]. In contrast, laminar recordings of epileptic activity in mesial temporal structures in patients have been more consistent with excitation in the deeper pyramidal cell layer[16,18,26]. From all of this work, it has been difficult to build a model explaining the physiology of epilepsy, particularly relative to cortical layers, likely since chemically-induced seizures within normal animal cortex are very different physiologically from what occurs in the human ictogenic cortex. However, one hypothesis emerging from these results is that there is differential involvement of the cortical layers during seizures, a hypothesis that could be tested in laminar recordings of spontaneous epileptic seizures in humans in vivo.

Here, we used laminar recordings of seizures occurring spontaneously in patients during intracranial exploration for presurgical evaluation of drug-resistant epilepsy. These laminar recordings allowed estimation of the synaptic activity (represented by local field potentials, LFPs) and somatic action potentials (multi-unit activity) within each of the six cortical layers. This approach makes it possible to describe the microcircuitry of the neocortex that underlies the generation and propagation of seizures. We observed that ictal discharges within the seizure onset zone came from the granular and infragranular layers, whereas the ictal discharges in areas outside of the onset zone came from the supra-granular layer.

## Results

### Patients
We recorded 30 seizures from 10 individuals (2 females, 8 males) who underwent video-EEG recordings (intracranial and scalp) as part of their evaluation for intractable focal epilepsy at four collaborating institutions (Massachusetts General Hospital & Brigham and Women's Hospital, Boston; New York University Medical Center, New York City; National Institute of Clinical Neurosciences, Budapest). Different etiologies accounted for epilepsy including focal cortical dysplasia, mesial temporal epilepsy, and tumour, in a single case (grade II oligodendroglioma). Similarly, different neocortical regions were affected (see Table S1). Seven out of the 10 patients had two laminar electrodes implanted and three only had one. Among the 17 laminar electrodes, 5 laminar electrode implants were considered as being in the ictal onset zone (also called the seizure onset zone), based on the simultaneous clinical recordings. Only 15 out of the 17 laminar electrode recordings were used in this study as too many artifacts were present on two electrode recordings (with both excluded electrodes outside the ictal onset zone). Moreover, one of the 15 remaining electrodes did not show any ictal activity (manual examination by the clinical team, reviewed by an independent expert) during the seizure despite a good quality signal, which was interpreted as the absence of seizure propagation to the region around this electrode.

### Discharge generation and propagation
After identifying the recordings from the ictal onset zone and those from the seizure propagation zone (Fig. 1), we identified each ictal discharge (see Methods). A total of 614,071 ictal discharges were identified across all electrode channels (408 channels) while 170,607 (27.8%) discharges were identified in the seizure onset zone. Current Source Density (CSD) measurement has been used to localize the source of the LFPs within the cortical layers and primarily reflects post-synaptic activity (see Methods)[15]. Briefly, the calculation of the CSD produces, at each time point, sinks and sources of current flow. Measuring CSD, then, makes it possible to locate the current flow between the cortical layers and therefore those which are involved at that time. We identified all the events corresponding to a single epileptic discharge and generated a normalized mean CSD across the cortical layers (Figs. 2, 3). During the generation of ictal discharges, the CSD values were higher in the supra-granular layer compared to deeper layers when recording outside the ictal onset area. Within the seizure onset area, the CSD was larger in the deeper infragranular and granular layers (chi-2, $p < 0.0001$; Fig. 2). It is worth mentioning that these observations are consistent with what has been described for inter-ictal discharges previously[27,28].

### Dynamics in circuit activity during seizure progression
These results show the difference in involvement of the cortical layers in the generation of discharges, depending on whether these are recorded directly in the ictal onset zone or in a zone of seizure propagation. However, since epileptic seizures are not static processes, we analyzed the evolution of those patterns during the seizure. Inside the seizure onset zone, the involvement is always in or below the granular layer with changes during the seizure occurring only in the exact pattern of sinks and sources across the cortical layers. In contrast, outside the seizure onset zone, the sinks and sources start above the granular layer and, while that same basic motif is conserved over time, with the initial sink of each ictal discharge engaging deeper and deeper layers as the seizure progresses.

Within the ictal onset zone, the CSD maxima and minima constituting the alternating sinks and sources remain in the infra-granular and granular layer throughout the seizure without any extension into the more superficial layers (Fig. 2A). None-the-less, the pattern details change in the early initiation compared to later phases of the seizure. Initially, the sink is observed both in the granular cortex / superficial infragranular cortex as well as in the deepest layers of the cortex, thus flanking a source in the middle part of the infragranular cortex. A few seconds after seizure initiation, the sink encompasses the entire granular cortex and the infra-granular cortex (at a very early period of the discharge generation) excluding the deepest portion of the infra-

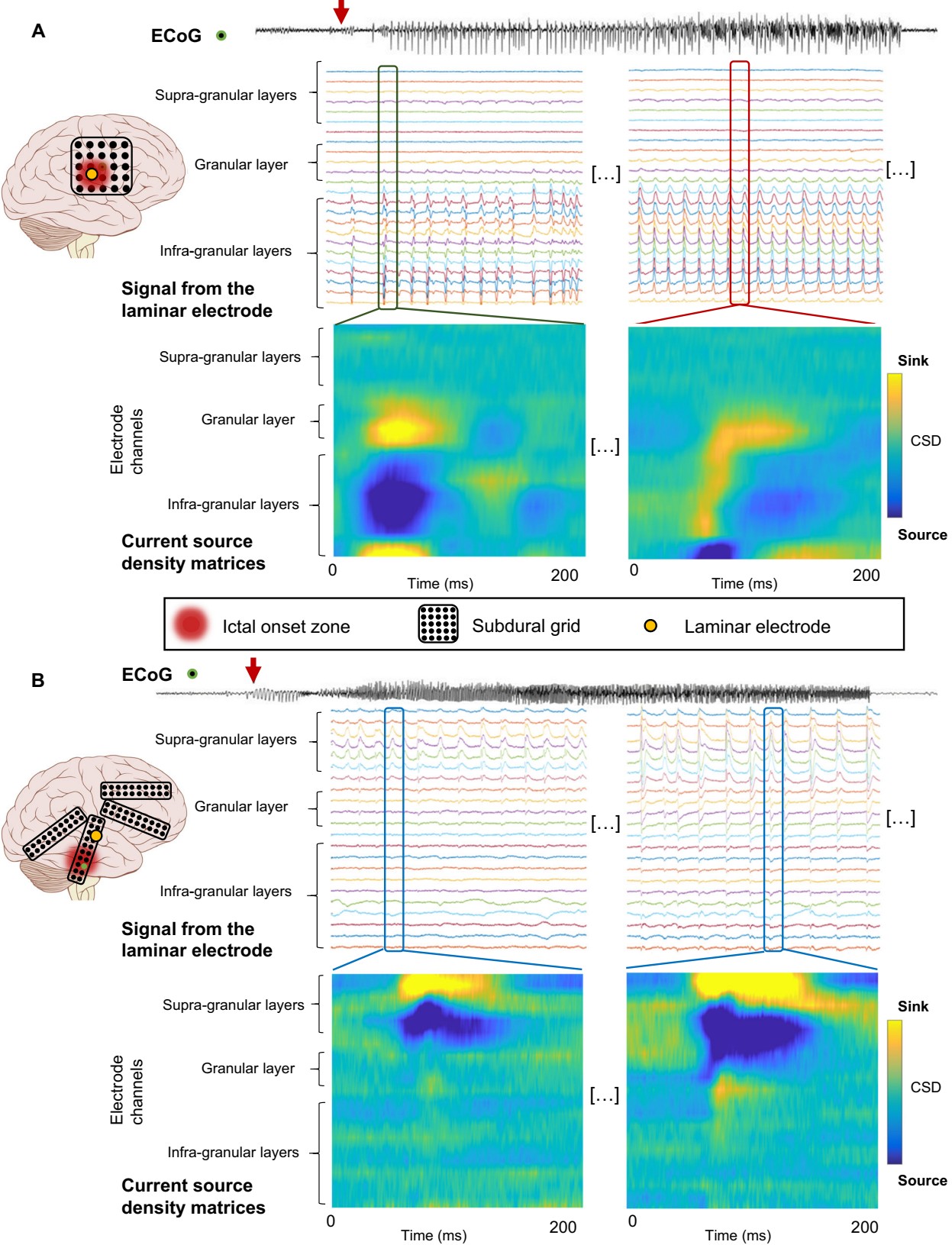

granular cortex which constitutes the source (Fig. 1, 2). Conversely, in the seizure propagation zones, the increase in the maximum values of the CSD seems to be confined to the supra-granular layers at the start of the seizure. This large superficial sink and slightly deeper source remain throughout the seizure in the propagation zone. However, the sink extends into deeper layers of the cortex as the seizure progresses.

(Fig. 2A). It is worth noting that the sink/source pattern does not change before seizure termination, even as ictal discharging becomes more intermittent.

To quantify the changing dynamics of laminar involvement more completely, we examined the ictal discharge current flow, particularly comparing and contrasting events in the seizure onset zone versus

**Fig. 1 | Example of LFP (Local Field Potential) and CSD (Current Source Density) inside and outside the seizure onset zone. A** The laminar electrode (yellow dot) was implanted within the ictal onset zone (red shading). Seizure onset (red arrow) on laminar recordings is similar to the adjacent ECoG electrodes. Ictal activities were first observed in granular and infra-granular layer recordings (sources are plotted in blue) with sinks both in the adjacent granular cortex and in the deepest infragranular cortex. Later during the seizure, the source moved into the deepest layer and the sink is constituted by all the remaining granular and infragranular cortex. **B** The laminar electrode (yellow dot) was implanted outside the ictal onset zone. Seizure onset (red arrow) on laminar recordings is late as compared to the adjacent ECoG electrodes. Ictal activities were first observed in supra-granular layers with sinks in the adjacent supra-granular cortex. Later during the seizure, the pattern appears to be more intense but grossly similar in morphological features.

outside the seizure onset zone. During the generation of ictal discharges, the CSD reaches its peak more frequently in the supra-granular layer in recordings outside the ictal onset area. Within the seizure onset area, the CSD reaches its maximum in infragranular and granular layers (chi-2, p < 0.0001; Fig. 3A). Again, these observations are consistent with what has been described for inter-ictal discharges previously[27,28]. Thus, we find supra-granular, granular, and infra-granular layers are not homogeneously involved in seizures. Inside the ictal onset zone, at the time of the ictal discharge generation, sinks and sources are simultaneously present in granular and infra-granular layers and noticeably alternated in time: each sink at the time of the generation of a discharge tends to become a source in the 50 ms following the discharge onset and vice versa. These oscillations between sink and source are not observed in the supra-granular layers. Similarly, in recordings made outside the ictal onset zone, this alternation of sink and source is also observed, but only in the supra-granular layers (Fig. 3B).

To quantify these observations across all 10 patients and 30 epileptic seizures, for each patient, both the variance and the normalized mean of all the positive (sinks, normalized between 0 and 1) and negative values (sources, normalized between −1 and 0) of the CSD were computed. We analysed 8 seizures in 4 patients in the ictal onset zone and 23 seizures in the propagation pathway in 8 individual patients (averaging within patients then across patients). The mean normalized CSD values were significantly higher (t-test, $p < 0.001$) within the ictal onset zone for the granular and infragranular electrodes compared to the supra-granular electrodes for both sources (−0.350+/− 0.212 vs −0.099+/− 0.098) and sinks (0.382+/− 0.249 vs 0.077+/− 0.066). Unsurprisingly, the higher values in the ictal onset zone were associated with significantly greater variance (t-test, $p < 0.005$) for both sources (0.033+/− 0.035 vs 0.235+/− 0.174) and sinks (0.039+/−0.051 vs 0.325+/− 0.202), reflecting the temporal alternation of sources and sinks in the granular and infragranular cortex within the ictal onset zone during epileptic seizures. In contrast, outside the seizure onset zone, the normalized mean CSD was significantly higher (t-test, $p < 0.001$) in the supra-granular cortex than in the other layers, both for sources (−0.507+/− 0.292 vs −0.197+/− 0.153) and sinks (0.503+/− 0.212 vs 0.112+/− 0.111). The significant difference in the analysis of variances (t-test, $p < 0.005$) between the supra-granular cortex and the other layers was in the same direction for sources (0.418+/− 0.217 vs 0.091+/− 0.097) and sinks (0.383+/− 0.221 vs 0.035+/− 0.067). The alternation of sources and sinks was indeed found outside the seizure onset zone when recording the propagation of seizures, though found only in the supra-granular layers of the cortex (Fig. 3B), at least at the initiation of the seizure.

To examine how these patterns of laminar involvement change over time quantitatively, we utilized two different approaches. First, we performed an independent component analysis on the CSD of individual ictal discharges in all patients. Within the ictal onset zone, we identified two distinct CSD patterns for discharge generation. The first pattern was found at the start of the seizure, with a discharge frequency starting at 1.3 Hz (+/− 0.4) during the first second of the seizure that increases to 8.9 Hz (+/− 2.1) at the end of the pattern, lasting on average during the first 5.3 s (+/−1.2). The first pattern was characterized by sinks at the deepest and most superficial level of the infragranular layers with a source in between. Immediately after ictal

discharge initiation, the most superficial sink and source reversed. Then, during the remainder of the seizure, the second pattern was identified lasting until the end of the seizure, lasting 57 s (+/− 12.4) on average, with a constant discharge frequency of 7.1 Hz (+/− 0.9) except for the end of the seizure where ictal activities became intermittent. This second pattern was characterized by a predominant sink and, for the duration of the pattern, in the most superficial part of the infragranular layers with a source at the deepest part. Between the two patterns, all the remaining infragranular layers were a sink transforming very quickly into a source (Fig. 4A). In contrast, the analysis of the discharges identified in the seizure propagation zone only allowed the identification of a single pattern using the same independent component analysis methods (Fig. 4B).

In a second analysis, to enable a temporal comparison of the evolution of this activity across seizures, we normalized the duration of all seizures to 100-time points. At each time point, the average absolute value (to include both sinks and sources) of the CSD was averaged for all seizures of a single patient and then across all patients. If the latter deviated by more than 2 standard deviations from the previously calculated base value, it was considered an outlier. The summation of all the resultant values made it possible to perform a normalized analysis of their distribution across the cortical layers over time.

Inside the ictal onset zone, no significant variation in the normalized CSD across all patients was found either within the supra-granular layers ($p = 0.391$) or within the infra-granular layers (p = 0.052, (Fig. 4B). Conversely, there was a significant increase (p < 0.01) of the normalized CSD within the granular layer between the first 24% of the seizure (mean = 0.075 + /− 0.042) and the later part (mean = 0.193 +/− −0.057).

Outside the ictal onset zone, in seizure propagation areas, we noted a significant increase in the normalized CSD across all data between the first 26% of the seizure and its continuation for the supra-granular layers ($p < 0.001$, mean = 0.610 + /− 0.041 vs mean 0.787 = +/− 0.069), after 28% for the granular layer, ($p < 0.001$, mean 0.008 = +/− 0.022 vs mean 0.406 = +/− 0.093) and after 42% for the infragranular layers ($p < 0.001$, mean = 0.021+/−0.016 vs mean 0.057 = +/− 0.044).

## Multi-unit activity during discharge generation and seizure propagation

The analysis of the LFPs and the CSD mainly reflect postsynaptic activity[29]. Another part of the circuit is represented by the multi-unit activity (MUA) which largely reflects local neuronal action potential firing. Of the 30 epileptic seizures analysed, the MUA could only be analysed in 22 cases; in 7 cases, signal quality in the MUA band was too poor for analysis and, in a final case, there was technical failure to record MUA during data acquisition. Within the ictal onset zone, there was an increase in MUA in the infra-granular layers of the cortex with a predominance in the deepest part of these infragranular layers during ictal discharges. In the seizure propagation zone, at the time of discharge generation, the increase in CSD in the most superficial parts of the supra-granular cortex was accompanied by multi-unit discharges in those same layers, as well as in the granular and most superficial part of the infragranular layers (Fig. 5A). To quantify these observations across all participants and seizures, we used a similar approach to the CSD analysis described previously. For each seizure

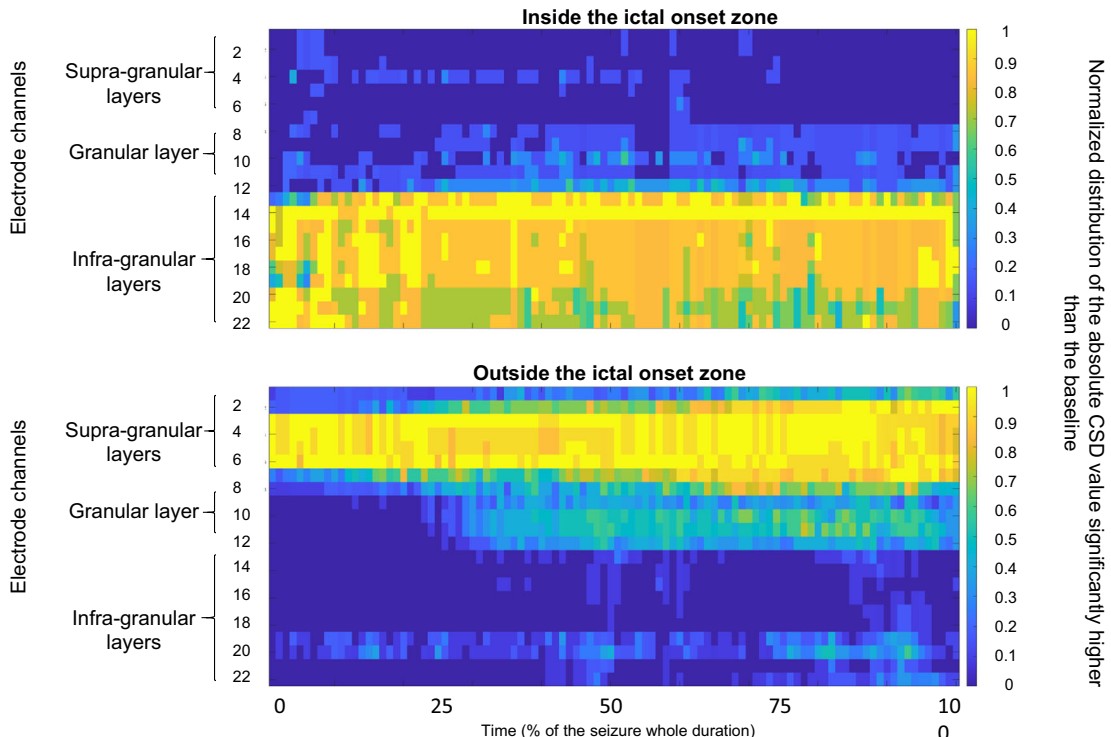

**Fig. 2 | Activity evolves through time across layers during a seizure. A** Example in a single participant having two laminar electrodes of change of the CSD over time for a seizure recorded within the ictal onset zone (top) and outside the ictal onset zone (bottom). The start and end of seizures are marked with red arrows. The red lines represent the CSD distribution across the cortex and the clear change in the CSD spread for seizures outside the onset zone. Insets in the middle show an expanded view of the CSD at different time points. **B** Average representation of CSD evolution across all participants and seizures during seizures recorded within the ictal onset zone (upper matrix) and outside of the ictal onset zone (bottom matrix). The values correspond to the normalized distribution of the absolute values of CSD (thus combining sink and source) differing significantly from the baseline.

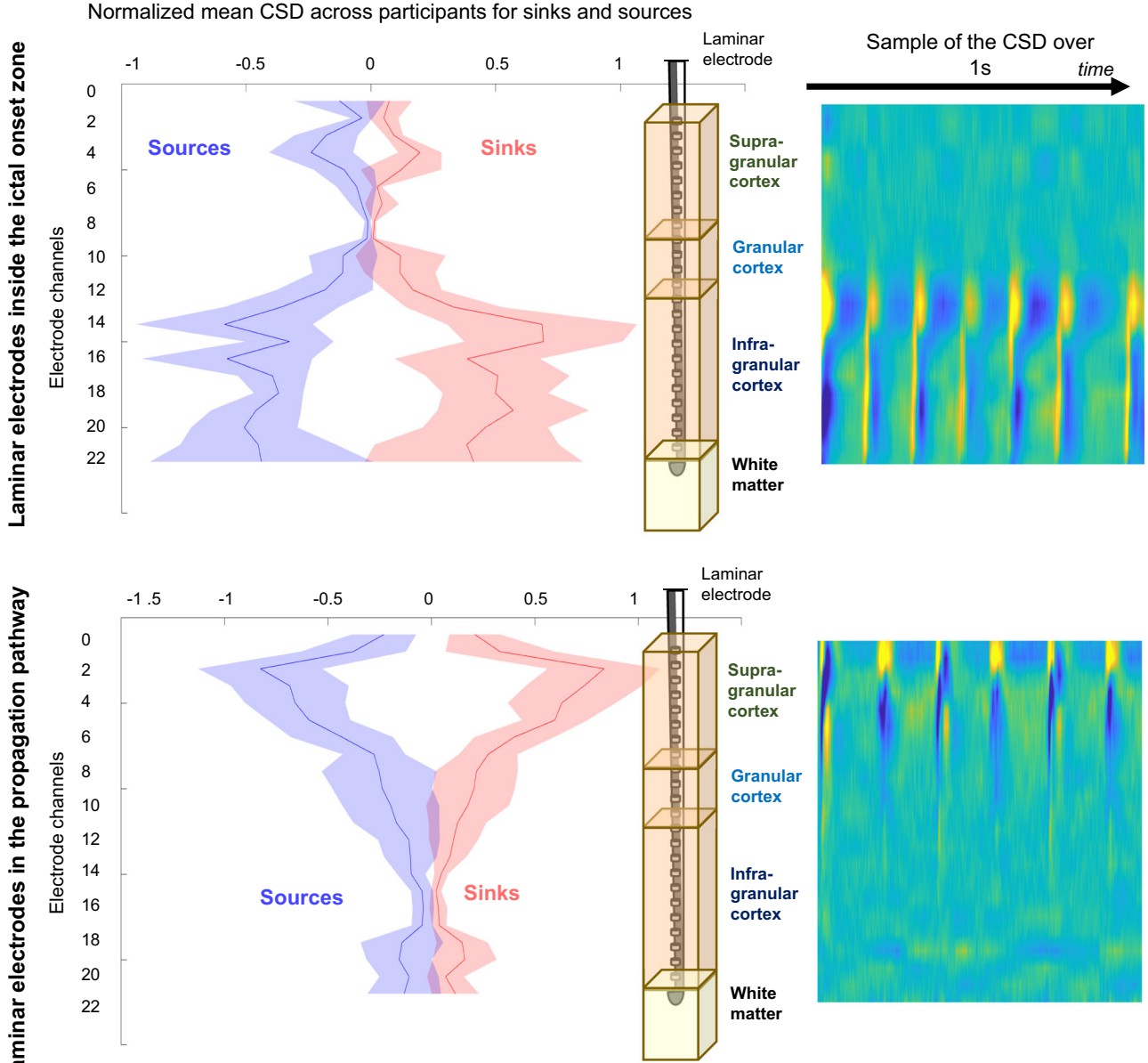

**Fig. 3 | Distribution of CSD values (mean values +/- SEM).** Left: Normalized mean across subjects and seizures of the CSD for sources (in blue) and sinks (in red) for electrodes implanted in the ictal onset zone (left) and in an area of spread (right). The CSD is not limited to the ictal discharges but is computed on the whole signal during the seizures. Right: Samples of CSD raw data with the same colour scale as for Fig. 1 and recorded in supra and infra-granular are represented to illustrate the temporal alternation of sources and sinks. Example from single participants of the sink/source alternation over 1 s during a seizure.

for which both CSD and MUA analysis were available, we performed an analysis of both CSD and MUA discharges after normalizing MUA between 0 and 1. We then compared these values averaged over time for all these data for each of the electrodes (Fig. 5B). Within the ictal onset zone, like the CSD, there was no significant modification of the MUA values within the supra-granular and granular layers compared to the baseline calculated over a 10 s pre-ictal period. In contrast, in infragranular-layers, there was a significant increase in MUA (0.398+/− 0.058 vs 0.083 +/− 0.006, t-test, $p < 0.001$). Remarkably, more than half of the MUA was due to the deepest 20% of channels which is in contrast with the more homogeneous change in CSD throughout the infragranular layers. In the propagation pathway, again largely mirroring what was observed for the CSD, there was a significant increase in MUA in supra-granular layers. More specifically, this

significant difference (t-test, $p < 0.001$) was due to the most superficial half of the supra-granular layers (0.307+/− 0.163 vs 0.032+/− 0.001). This increase in neuronal firing reflected in the MUA was also seen in the granular layer (0.116 +/- vs 0.032+/− 0.001, t-test, $p < 0.005$), as well as the most superficial half of the infragranular layers (0.071 +/− vs. 0.032+/− 0.001, t-test, $p < 0.005$).

## Discussion

We used laminar microelectrode arrays to obtain unique information about the cortical grey matter circuit elements involved in seizure generation and propagation. Specifically, we show that the initial generation of seizure activity exclusively involves the granular and infragranular cortical layers. These results imply that intrinsic attributes of deeper neocortical structures are crucial for the seizure

**Fig. 4 | Independent Components Analysis (ICA) reveal evolving patterns differentiating the ictal zone from outside the ictal zone. A** recording from the ictal onset zone. **B** a propagation region. In both **A** and **B**, the top plot shows the CSD over time in all channels. The middle plot shows the results of the independent component analysis during the same time period, and the lower plot represents the identified pattern(s). A diagram of the temporal dynamics of detected patterns is illustrated below the lower plots in both **A** and **B**.

initiation process. Within the ictal onset zone, in the first seconds following the start of the seizure, infragranular MUA is observed at the time of discharge generation, the most intense activity in presumed layer VI. Concomitantly, the CSD analysis shows current flow only in the granular layer and the deepest portion of the infragranular cortex. The layer IV current sinks could reflect synaptic activity in proximal portions of the apical dendrites of layers V and VI pyramidal neurons, and we hypothesize that a pyramido-pyramidal intrinsic excitation loop drives this current flow and unit activity.

Following the short seizure initiation phase, the activity within the ictal onset zone changes. During the initial phase, the MUA is confined to the infragranular cortex. However, current flows in the granular layer are even more important and those towards the infra-granular cortex are reaching layers IV and V. Despite these modifications, it should be noted that throughout the development of the seizure within the ictal onset zone, there is no current flow to supra-granular cortex, which is surprising considering the canonical organization of the neocortex in which normal pyramidal cells from the infragranular and granular layers

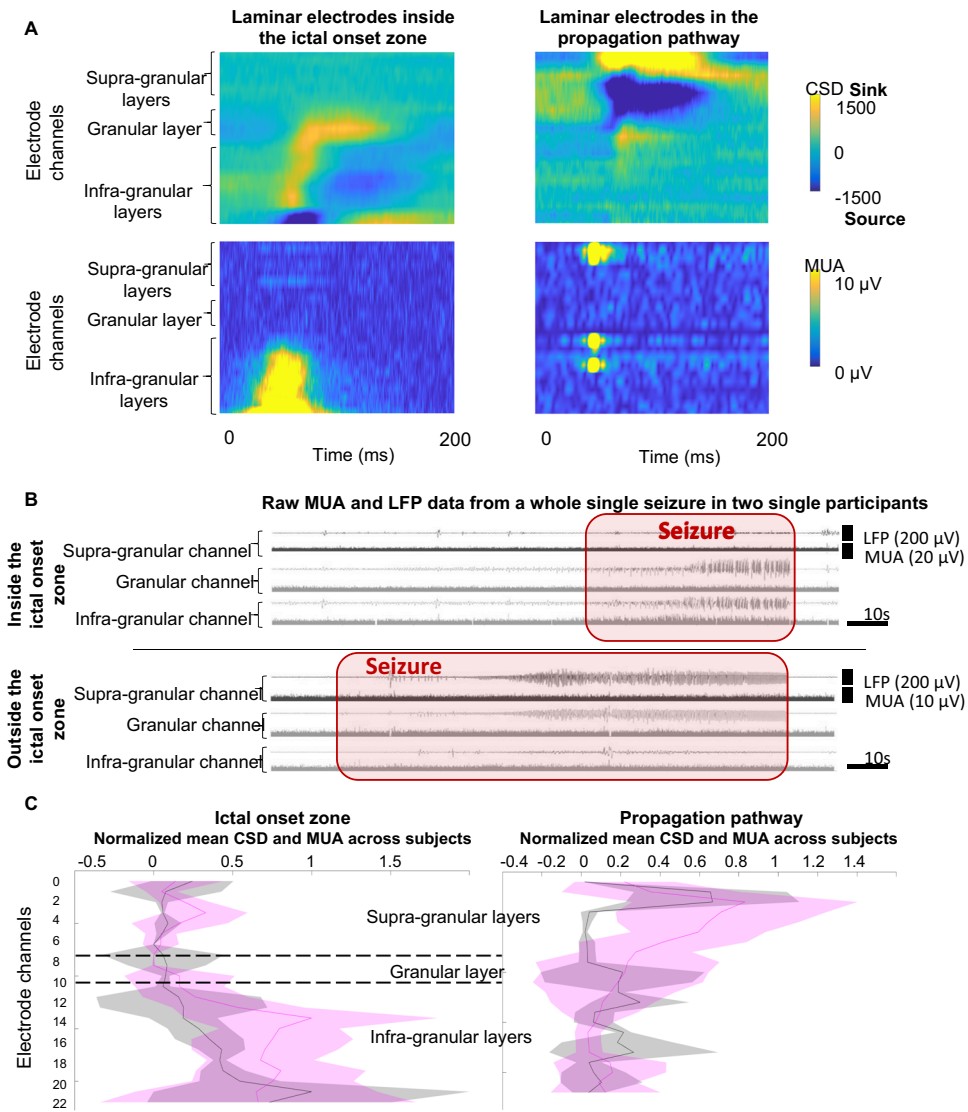

**Fig. 5 | Multi-unit activity (MUA) changes across cortical layers alongside CSD.**
**A** Example of CSD (top) and MUA (bottom) during ictal discharge generation inside the ictal onset zone (left) and the propagation pathway (right). **B** Example of three single laminar channels from supra-granular, granular and infra-granular layers from two participants showing both the LFP and the MUA during the whole seizure.

Top, laminar electrode implanted inside the ictal onset zone. Bottom, laminar electrode implanted outside the ictal onset zone. **C** Normalized mean (mean values +/- SEM) CSD (purple) and MUA (black) across participants (*n* = 9) inside (left) and outside (right) the ictal onset zone (*n* = 22 seizures).

send massive projections to supra-granular cortex[30–32]. Similarly, changes in MUA are confined to granular and infragranular regions. The absence of significant changes in MUA in the supra-granular cortex during this period suggests that the more superficial cortex is not involved. What could explain the functional disconnection between superficial and deep cortical layers? One hypothesis is that pathological neurons of the ictal onset zone have dominant recurrent collaterals within layers V and VI projecting onto dendrites and cell bodies within infragranular layers (Fig. 6). This pathological circuitry could also be associated with a feedforward inhibition through projections of the layer V and VI pyramidal cells onto interneurons responsible for inhibition of the layer II and III pyramidal cells.

Among these observations, the activity within the deepest layer, layer VI, is unique. Indeed, during the first seconds of seizure initiation, layer VI behaves similarly to the most superficial portion of the infra-granular layer with increased neural firing as represented by MUA. However, after seizure initiation, the activity of this layer differs from the rest of the entire cortex by becoming the predominant source of current during ictal discharge generation. The sink-source alternations

therefore seem to be opposite in relation to the rest of the infra-granular cortex. If we cannot exclude an artefactual origin related to the junction between the cortex and the white matter, it is also possible to make a connection with the very particular structure of layer VI in human cortex. This primate-specific layer had been noted from the first histological descriptions of the human cortex[33,34], but the fine cytoarchitectonic and functional description of the projections is more recent[35,36] and suggests specific excitatory projections of triangular cells on layers III and IV. Even more recently, triangular cells specific to layer VI have been identified as having large horizontal dendritic projections within this layer[32]. The latter could play a role in the singular behaviour of this deep layer. Unfortunately, to our knowledge, no laminar non-human data has been reported describing layer VI activity during seizures, which would be very useful for comparison with the data presented here since the architecture described seems specific to humans.

The spatial spread of the seizure along the epileptic network appears to utilize supra-granular layers, particularly the most superficial layers of the cortex. This general pattern does not vary during the seizure with significant current flow in layers I-III. Secondarily and to a

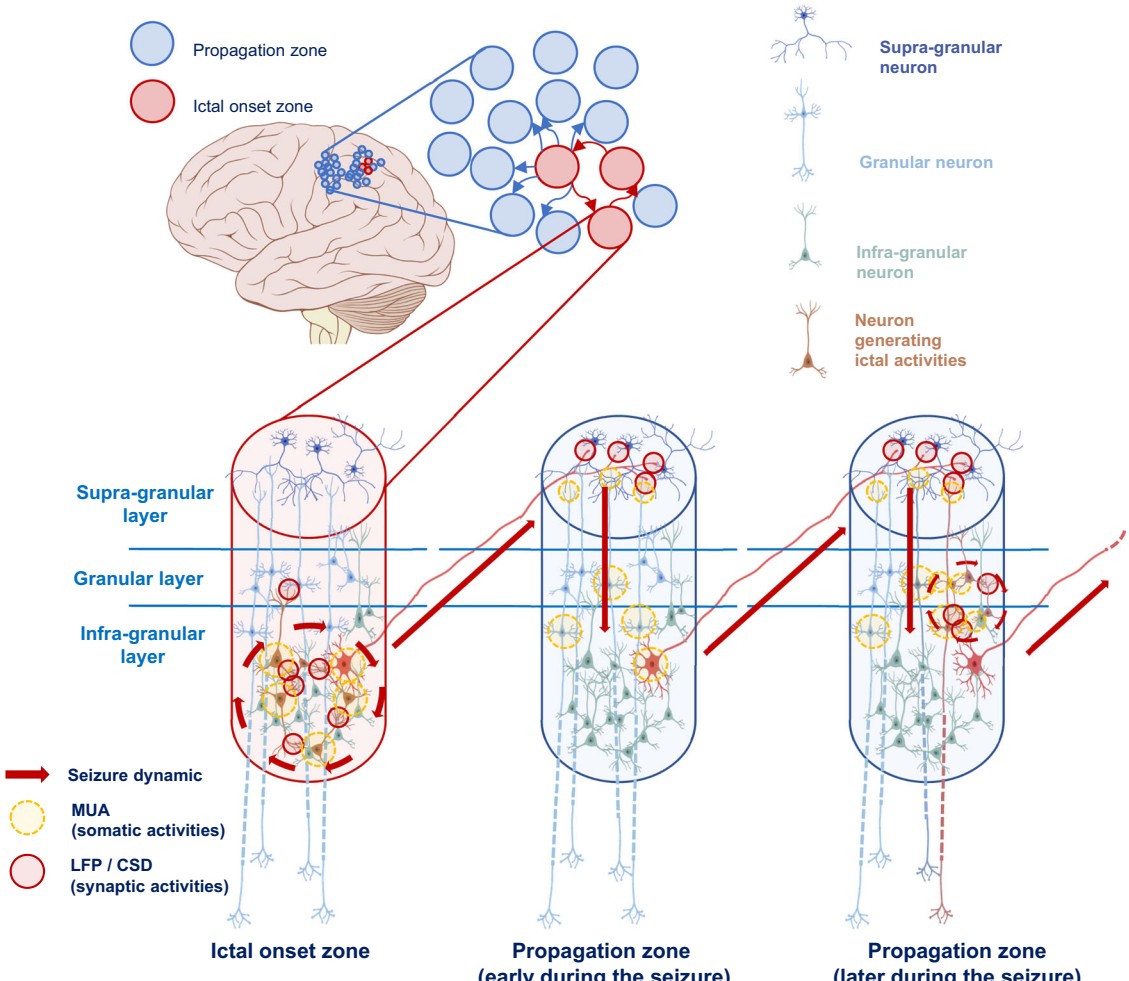

**Fig. 6 | Hypothetical schema of seizure generation in the ictal onset zone (red column) and its propagation (blue columns).** Red arrows show interpopulation spread of ictal activity. The exact cell types involved in each layer remain to be described. The red circles represent the regions where a pathological CSD has been demonstrated and the yellow ones where an increase in MUA has been described (see Fig. 3).

lesser extent, seizure activity-related current flow involving the deeper layers becomes notable. This result seems, at first, counterintuitive insofar as layer I is mainly composed of interneurons to which the pyramidal neurons of the deepest layers of the cortex, and in particular layer IV, project[32,37,38]. These results suggests that there are direct projections to superficial cortical layers in the propagation pathway. Activation of synapses in this pathway leads to current sinks and sources in apical dendrites localized in superficial layers as well as activation of soma in superficial areas of supragranular cortex. Later in the seizure this activity remains but there is also activation of synapses in layers III and IV perhaps through collaterals which project to proximal pyramidal cell dendrites. This deeper layer current later in the seizure might result in "autonomization" within the seizure propagation zone in which that local area is now generating seizure activity of its own rather than simply reflecting propagation from the primary onset area.

The MUA data also supports the infragranular origin of the ictal discharges within the seizure onset zone, while no modulation is noted in the more superficial layers. This contrasts with the results obtained with the Utah arrays whose total depth is 1–1.5 mm and are likely recording from Layer IV and above[39,40]. In those studies, single and multi-unit neuronal activity are observed at the start of the seizure in presumed Layer IV and above. The first explanation for this discrepancy relates to the intrinsic characteristics of the electrodes. The Utah arrays with their fine-tipped electrodes may be more sensitive than the laminar microelectrodes to single and multiunit activity. In addition, the Utah array may be able to record such activity at a greater distance than the laminar electrodes and thus be able to record from deeper layers even if those regions are not directly implanted by the array. Another explanation could be based on the exact location of the electrodes. Despite an electrode designed to ensure orthogonal penetration to the cortex, the relationship of the laminar microelectrode array to specific layers remains an indirect deduction and represents a limit of this study.

A recent study examining coupling between sub-dural recordings and 1.5 mm deep Utah Array[41] microelectrode recordings showed that within-ictal onset zone activity behaves like a dynamic zone. The researchers propose an interesting dynamic model in which seizures evolve into self-organized structures wherein a small seizing territory projects high-intensity electrical signals over broad cortical areas. These results highlight the importance of interactions with other distant and favourable zones in the progression of the seizure. These results in Utah array recordings may at first appear contradictory with the laminar electrode data reported here where we communicate fairly static epileptic focus and spread zones during the seizures. However, further comparisons between our study and the Utah array study from Smith et al.[41] demonstrate there is convergence. First, the Utah array type electrodes used in this study allow the recording of MUA and LFPs in a granular layer through many recording sites. Thus, the Utah array MUA is of better quality (more abundant data enabling cell type

identification) but only records at one depth of the 6 cortical layers. Conversely, the laminar electrodes used in our work may have fewer recording channels (with MUA more difficult to analyse), but can record each of the 6 cortical layers. In the results from the Utah array study, there is a period of 5 to 10 s (see Fig. 5a in Smith et al.)[41] between pre-recruitment and the end of the ictal wavefront preceding a very homogeneous post-recruitment and pretermination periods in terms of MUA. This characteristic initial phase in MUA found in the Utah array recordings could therefore correspond to the pattern described here in the first seconds of seizure initiation (Fig. 4) and precede the homogeneous activity observed throughout the seizure. The appearance of a static phenomenon in our reported data would come from the uniqueness of the laminar recording site within the epileptic focus. We hypothesize that the wavefront as reported in the Utah array recordings could have been observed if several laminar electrodes had been put within the ictal core, which may be of interest in future work. However, regarding the electrodes placed in the propagation pathways, we further hypothesize that the contacts were probably too distant from the ictal onset zone to participate in seizure generation and only became autonomously active very late when the deeper layers were recruited leading to non-propagated local discharge generation.

This first analysis of human laminar data during seizures concentrated on a relatively homogenous population of seizures with clear focal onset and spread. There does not appear to be a relationship between aetiology and the mechanisms observed during seizure generation. This lack of a relationship, though, provides an argument in favor of the existence of generic mechanisms that might underlie complex focal epileptic seizures. Nonetheless, it is important to recognize that seizures with different onset patterns, particularly seizures with a more diffuse and generalized onset, often seen in the most difficult-to-control epilepsies, may present a different pattern of cortical laminar activity.

In terms of the therapeutic implications of these results, a confirmation of a specific signature of the seizure onset zone and its identification on inter-ictal recordings could lead to the development of laminar electrophysiology as a tool for intraoperative identification of the region to be removed during surgical resections of epileptic foci. Moreover, if the development of hybrid stereoelectroencephalography (SEEG) electrodes including a laminar component between the macroscopic recording sites continues, the identification of this signature would greatly facilitate the interpretation of SEEG. Indeed, this may make it possible to identify when sEEG sampling has failed to identify the seizure onset zone by ensuring that earliest involved region has infragranular (locally generated) ictal discharges. Furthermore, this approach could better identify multifocal epilepsies by identifying infragranular origins in several distinct regions. Ultimately, this signature could be used in the design of close-loop devices that identify the seizure onset zone and the propagation pathways and prevent generation/propagation by stimulation, cooling or drug delivery. Finally, if a certain population of pyramidal cells were identified as responsible for the seizure initiation, those cells could be targeted pharmacologically leading to better seizure suppression versus side effects.

In conclusion, the results of this study show microcircuitry specific to the ictal onset zone involves only the deeper layers of the cortex suggesting that seizure initiation capacity is characterized by infragranular activity which remains restricted during the course of the seizure. Conversely, seizure spread involves a relay to the granular and upper infragranular layers for the entry of epileptic activities. As the seizure progresses, there is greater involvement of deeper layers in the region of propagation, even while the seizure onset zone continues to involve only the deepest structures. This asymmetry, regardless of the identified aetiology, suggests that there are generic mechanisms underlying the ability of an area of the cortex to generate and induce propagation of seizures and that the physiological signatures of these

mechanisms may be useful as a biomarker for seizure onset areas and offer novel approaches for seizure management.

## Methods

This research complies with all relevant ethical regulations: Partners Health Care IRB (currently Massachusetts General Brigham IRB), the New York University Medical Center IRB, and the Hungarian Medical Scientific Council.

### Patients and recordings

Patients presenting with drug-resistant focal epilepsies and requiring a phase 2 assessment as part of a pre-surgical evaluation were offered to participate in the study and to have a long-lasting intracranial electrophysiological recording with one or two laminar electrodes in addition to the electrodes used in clinical practice (Table S1). Patients were included after agreeing to participate in the study. All participants were informed that their refusal would not affect their clinical care. Experiments were made with fully informed consent and were approved by local institutional review boards (IRBs). These boards included the Partners Health Care IRB (currently Massachusetts General Brigham IRB), the New York University Medical Center IRB, and the Hungarian Medical Scientific Council.

The implantation of the electrodes for clinical purposes was not conditioned or influenced by the study. Patients were implanted with subdural grid electrodes (Ad-Tech Medical Instrument Corporation, Racine, WI, USA) and the recordings were continuously obtained (ranging from 10 to 14 days in the 10 participants) at a sampling rate of 256 or 512 Hz. The implantation of the laminar electrodes was done in areas suspected of being the ictal onset zone (cortex likely to be resected) based on phase 1 investigation including clinical examination of seizures, long-lasting video scalp EEG, high-resolution MRI, magnetoencephalography, and metabolic imaging. When technically possible, a second laminar electrode was implanted at a distance from the first electrode (always further than 5 cm, see Table S1). At the end of the intracranial recording, the laminar electrode(s) were explanted during the clinical electrode removal procedure.

At the end of the intracranial recording, all the clinical intracranial electrophysiological data were analysed by the epileptology team. First of all, the neurophysiologist made sure that nothing could suggest that the ictal onset zone is located outside the field of exploration of the subdural grid. If this was the case, the epileptogenic zone was then identified and it was possible to determine whether the laminar electrodes were located inside or outside the ictal onset zone.

### Laminar microelectrodes

Laminar electrodes arrays were 350 μm diameter, 24 contacts experimental multichannel microelectrode array electrodes. Each contact was Platinum/Iridium and 40 mm diameter spaced evenly at 150 μm. LFP recordings were thus recorded on 3.5 mm, spanning from layer I to layer VI[15–18,27,28,42–46]. A silicone sheet attached to the top of the microelectrode array shank prevented the first contact from sliding more than 100 μm below the pial surface. Differential recordings were made from each pair of successive contacts to establish a potential gradient across the recording sites. After wideband filtering (DC-10KHz) and preamplification (gain 10x, CMRR 90db, input impedance 1012 ohms), the signal was split into a low frequency field potential band (filtered at 0.2–500 Hz, gain 1000x, digitized at 2KHz, 16 bit) and a high frequency multiunit/single unit activity band (zero-phase digital high pass filtering above 300 Hz, 48 dB/oct, gain 1000x, digitized at 20KHz, 12 bit), and stored continuously. These laminar electrodes were implanted perpendicular to the cortical surface, their ends being thus positioned under the subdural grid[15]. Stimulus markers (i.e. electrical pulse generated artificially and sent simultaneously to the two EEG amplifiers) were used to synchronize the recordings from the multielectrode array and clinical macroelectrodes.

Post-operative CT scans coregistered with preopertive MRI made it possible to obtain the position of subdural grid electrodes and the relative location of the laminar microelectrode[47]. As cortical layers cannot be visualized on MRI, the position of each laminar electrode contact was determined in relation to the theoretical thickness of the neo-cortex and its depth from the pial surface and based on the observation made on post-resection examinations of the cortex when available.

### Signal processing

The low-frequency field potential signal was preprocessed with the FieldTrip[48] toolbox for MATLAB R2017a v9.2.0.556344 (Mathworks, Natick, Massachusetts, USA). Artifact rejection was performed manually using the ft_databrowser function.

The LFP and MUA were recorded simultaneously at 2000 and 20,000 Hz. A filter was then applied online from 0.2 to 500 Hz and 200 to 5000 Hz. Data from a channel was considered rejected after visual inspection of data were linearly interpolated from the channels directly above and below them (12.5% of channels). A 2 Hz bandwidth filter was applied at 50 Hz or 60 Hz according to the geographic origin of the recording to eliminate the line noise.

CSD was measured by taking the second spatial derivative of the LFP and then applying a 5-point Hamming filter[15,49,50]. Concerning the second-most deep and superficial channels of the laminar probe, CSD was estimated by using the Vaknin approximation which consisted of adding pseudo-channels of zeros to the LFP above and below the array[51].

Normalization was performed to make inter-participant comparisons possible. The equation for rescaling data X to an arbitrary interval [0 1] was: $X rescaled = 1 + \left[\frac{X - \min x}{\max x - \min x}\right](0 - 1)$.

An automated ictal discharge detection was used to locate the discharges and measure the CSD at the time of the discharges as well as the average of the CSD over a period of 100 ms, centred by the discharge. The automatic discharge detection was based on a previously described method (https://github.com/Mark-Kramer/Spike-Ripple-Detector-Method/commits?author=Mark-Kramer)[52]. The discharge detection was performed directly on the laminar recordings. Potential gradient modification was thresholded, and the speed of its modification (ascending and descending) was taken into account. A minimum event duration was also an adjustable parameter so as not to wrongly include artefacts. A visual control was carried out on a sample of 100 discharges for each patient to obtain a sensitivity and a specificity greater than 90%. To morphologically analyse the CSD pattern through the pads of the laminar electrode at the time of the generation of a discharge, we used independent component analysis in order to separate mixed signals (blind source separation)[53]. After manual thresholding of the scale, matrices corresponding to 1 s of recording and representing the CSD values over time for each electrode contact were produced and made it possible to construct time-lag representations. We used 8 components in our ICA analysis. Computations were then performed using FastICA under Python Jupyter (scikit). The relevant components were then identified and thresholded manually, which made it possible to identify whether the discharges had a unique generation pattern in terms of CSD and how, if these patterns were multiple, they evolved over time.

### Reporting summary

Further information on research design is available in the Nature Portfolio Reporting Summary linked to this article.

## Data availability

The data used in this work are available upon request (contact SCASH@mgh.harvard.edu).

## Code availability

The code used in this work are available upon request (contact SCASH@mgh.harvard.edu).

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

## Acknowledgements

Dr. Mark Kramer from Boston University for the discharge detection methods contributing to data analysis routine (NIH R01NS110669). The Monahan Foundation, the Fyssen Foundation, the Philipp Foundation, the Edmonds de Rothschild Foundation and the French National Society for Neurosurgery for PB; K24 NS088568, and R01 NS062092 for SSC, and DOD / CDMRP for PS;, Hungarian Brain Research Program Grant (NAP2022-I-2/2022), Hungarian Pharmaceutical Research and Development Laboratory Project (PharmaLab, RRF-2.3.1-21-2022-00015) for IU; ANR-17-EURE-0017 to J.R.K. for his work at PSL for JRK. Figure 6 was partially created with BioRender.com and Figs. 1, 6 use an adapted CC material from Patrick J. Lynch, medical illustrator (Creative Commons Attribution 2.5 License 2006).

## Author contributions

Pierre Bourdillon: analyses and code implementation; data processing; manuscript writing; figure. Liankun Ren: methodology and data processing. Mila Halgren: method development. Angelique C. Paulk: data processing, manuscript proofreading, figure correction. Pariya Salami: analysis design. István Ulbert: electrode design and data acquisition. Dániel Fabó: electrode design and data acquisition. Jean-Rémi King: coding of the independent component analysis. Kane M. Sjoberg: data processing. Emad N. Eskandar: data acquisition. Joseph R. Madsen: data acquisition. Eric Halgren: analysis design; manuscript proofreading. Sydney S. Cash: research direction, manuscript review, figure correction.

## Competing interests

The authors declare no competing interests.
