## [Peer Review File · Nature Communications]

Differential cortical layer engagement during seizure initiation and spread in humansREVIEWER COMMENTS

Reviewer #1 (Remarks to the Author):

Analyzing combined recordings of linear depth electrode and surface ECoG strips/grids in epilepsy patients during pre-surgical evaluation, Bourdillon and colleagues provide strong evidence for differential ictal spike dynamics across cortical layers within the presumed seizure onset zone versus propagation territories. The findings bear significant clinical value for pre-surgical diagnostic work-up in pharmacoresistent focal epilepsies, and substantiate our understanding of ictal spike generation and propagation in the human cortex. In sum, the dual laminar electrode recordings are highly appreciated, and the paper's main finding is a valuable addition to the field. There are a few things that could/should be improved.

1) General layout/narrative of the presentation of the study: when reading the manuscript in its current form, as a reader that is familiar with the matter, it felt immediately odd to look at average spike analysis across entire seizures. It's long known that seizures, and their grapho-elements, undergo an evolution throughout the seizure. Initial ictal spikes do morphologically look different to late ictal spikes. Being unsatisfied with the immediate analytical over-simplification from the get-go of the paper, I had made a bunch of comments already until I reached figure 4, where what is intuitive was now acknowledged, shown, and further analyzed. In my view, nothing speaks against having figure 4 as figure 1, as it contains everything that is shown in figure 1 and also parts of figure 2. If the paper and figure display were trimmed/re-arranged, this would provide more clarity from the beginning of the paper, and reduce redundancy of displayed CSD plots across the paper. The average analysis can still be made, as the authors find, in part, surprisingly stable CSD footprints, but the dynamic picture should be put on the table early on. Further, figure 4 has a SOZ and propagation laminar electrode from a single patient, which is the intuitive way of presentation for the reader (figure 1 shows separate patients having just one implanted laminar electrode each, which is not intuitive, if the majority of patients had two laminar electrodes, and the main interest of the paper is to contrast SOZ and propagation areas).

2) In the introduction, the authors give a brief, underwhelming overview of previous literature on the topic. Two of 3 referenced papers are in vitro studies, some in neonatal brain tissue, whereas other highly relevant in vitro work, and high-resolution in vivo studies on laminar ictal dynamics are not mentioned, although they exist. E.g. in vitro, see e.g. Telfeian and Connors, 1998, *Epilepsia* 39, 700–708). E.g. in vivo, there have been microprism-assisted cellular scale in vivo imaging studies in mice on layer-specific dynamics during seizure initiation and spread. These studies are relevant to the study presented here, the contextualization of the presented results, and the final schematic outline. Aeed et al. (*Ann Neurol* 2020;87:97–115) looked at seizure initiation, and found different dynamics to the ones presented here (this may however simply be due to the superficial local application of 4-AP), while Wenzel et al. (*Cell Rep* 2017;19:2681–2693) looked at layer-specific ictal dynamics in propagation areas in primarily unaffected cortex, and the results go in a similar direction as is reported here. Layer 4 as the entry point/leading edge during seizure propagation is also discussed there, although technical restrictions prohibited specifically addressing this possibility.

3) Unit activity: Could the authors distinguish between pyramidal and interneuronal units, or was this not possible, e.g. due to unit quality? This would be powerful for the contextualization of the area-specific LFP and CSD analysis performed here (e.g. fast switches between sinks and sources), and link it to cell-type specific firing dynamics. This way, one could also contextualize previous literature on cell-type specific firing dynamics during seizures within the SOZ and propagation territories, which would possibly substantiate the authors' findings of specific LFP/CSD dynamics as a biomarker for delineating SOZ and propagation area. As unit data was obtained, it would be fantastic, if a distinction between excitatory and inhibitory units could be made.

4) Recent studies (e.g. Smith et al., *Nat Commun*, 2016, doi: 10.1038/ncomms11098) suggest that the "focus", or the "ictal core" is in fact non-stationary during seizures. So, what could be considered the "focus", may no longer be when the seizure progresses in time and space. The results reported here give a rather "static" view of the focus. As seizures are known to be highly

dynamic in space and time, this is a bit surprising. Could the authors discuss this in more detail? Could some of the changes of laminar dynamics, e.g. in propagation area, be due to a switch from "penumbra" to "ictal core"? This could, again, be substantiated e.g. by looking at pyramidal and interneuronal unit firing.

5) Methods, page 21: <<[...] When technically possible, a second laminar electrode was implanted at a distance from the first electrode. [...]>>

Please provide the concrete distances. This is important, as e.g. the propagation area may not be just one entity, but consist of distance-dependent sub-parts (e.g. penumbra or distant propagation area).

6) Results, page 4: <<[...] The CSD was maximal in the supra-granular cortex for 51,949 of these events and in the granular or infragranular cortex for the remaining 118,658 [...]>>

Is it the optimal analysis to use max values here? If one is looking for "presence" of significant changes in sources/sinks, shouldn't this be specified with regards to interictal, or interspike interval during ictus? When using the max only, it could exclude layers that show significant changes in current flow as compared to ref. This may become especially problematic when then the spatial distribution of exclusively max CSD values is compared, as is actually done in figure 2.

7) Results, page 4: <<[...] Among the 17 laminar electrodes, 5 laminar electrode implants were considered as being in the ictal onset zone (also called the seizure onset zone), based on the clinical recording. [...]>>

Could this be contextualized in a bit more differentiated fashion, with one or two sentences? The SOZ is an inherently ill-defined zone, as one would otherwise not need intracranial electrode implantations to further delineate the factual SOZ. As a necessary consequence, and as opposed to animal research, where this can be much better controlled, one cannot be certain about whether an implanted electrode really sits where the seizure starts, or ictal spikes are generated. The placement is based on the pre-surgical diagnostic data at hand, plus an "educated guess". The laminar electrodes have, due to their linearity, minimal x/y coverage (if the depth direction is considered as z). Thus, it's a realistic scenario that a laminar electrode that is implanted "inside" the SOZ, is in fact not. It may simply be recruited early on, as it is close-by the SOZ. This does not make the findings shown here less relevant by any means, but it's generally unclear whether there are just two, or more distinct ictal compartments (e.g. SOZ, nearby prop, distant prop). In how many cases were the areas, where the "SOZ laminar electrodes" were implanted, factually surgically removed subsequently, with good outcome? This would be a posthoc prove that the electrodes had indeed been situated in the SOZ, and it would be a relevant information to put into this paper, as it would indeed strengthen the usefulness of the findings in a biomarker context, especially for patients with pharmaco-resistant "non-lesional" focal epilepsies.

8) Results, page 8: <<[...] We analysed 8 seizures in 4 patients in the ictal onset zone and the recording of 23 seizures in the propagation pathway of 8 individual patients [...]>>

Does that mean that in 3 patients, no seizure was recorded in the SOZ although they had two laminar electrodes? Does that mean that the intra-SOZ laminar electrodes were actually not in the SOZ in 3 patients? How were these electrodes analytically dealt with (spatial re-assignment)? This also refers back to the general issue of the ill-defined SOZ.

9) At times, the readers have to reconstruct some of the numbers/aspects themselves, which disturbs the reading process unnecessarily. Simply write it out, please, e.g.:

Results, page 4: <<[...] Seven out of the 10 patient had two laminar electrodes implanted. [...]>>

The three remaining patients could have each 1 or >2 electrodes. Only the next sentence indirectly reveals that it must be a single electrode each for the remaining 3 patients. Suggestion: Seven out of the 10 patient had two laminar electrodes implanted, three patients a single laminar electrode.

Or results, page 4: <<[...] The CSD was maximal in the supra-granular cortex for 51,949 of these events and in the granular or infragranular cortex for the remaining 118,658 [...]>>

Please make it easier for the reader by e.g. adding "Inside the presumed onset zone," at the beginning of the sentence. The sentence before does not talk about inside/outside, and the sentence before that one mentions both, so the reader has to add up the numbers themselves (in looking at the following sentence where "Outside the ictal onset" is specified), to find out that the authors are talking about SOZ here.

10) Grammar/Typos/misspelled words; e.g.:

- Cover page: National Center for Neurological Disorders, , Xuanwu Hospital, Clinical
- Page 1: These latter electrodes an have multiple form factors
- Results, page 4: Seven out of the 10 patient had two laminar electrodes
- Methods: evaluation were offered the participate
- Table 1: Hippocampic
- Methods: LFP recordings was thus recorded on 3.5 mm
- Methods: The LFP and MUA were recorded simultaneously at 2000and 20000 Hz.
- Figure 1, legend: <>

Maybe I am mistaken here, but shouldn't it read "granular" cortex here?

- Figure 2, legend: <<[...] for electrodes implanted in the ictal onset zone (left) and in an area of spread (right). [...]>>

Left and right and top and bottom are not correctly labeled here I assume.

11) Figure legend, figure 1: Please specify red arrow; this is misleading with the red shading of the SOZ just above, in the figure. Of course, the arrow is temporal, not spatial, but still, some readers may be confused briefly.

12) "Spike generation" is numerous times across the paper, at times in a way that may trigger confusion or misunderstandings in the reader, please re-word in unambiguous/clear terms. Some examples:

- Results, page 8: <<[...] To better understand how spike generation differs within and outside ictal onset zones at the cortical level [...]>>

"Spike generation" is intuitively tied to a point of generation from where it then spreads. The wording here suggests an innumerable number of generators located inside and outside the SOZ. Possible solution: "How spike dynamics differ within" and so on.

- Results, page 4: <<[...] During the generation of ictal spikes, the CSD reached its peak more frequently in supra-granular layer when recording outside the ictal onset area. [...]>>
Each spike is inherently generated. Here, the authors are not analyzing spike generation per se, but simply the CSD during, or across individual spikes in the presumed SOZ and outside of it.

Reviewer #2 (Remarks to the Author):

This is an excellent and essential paper to understand seizure generation in human cortex. I have a few comments. Fig 2; does the "ictal spikes" refer to pre-seizure epileptic sharps. If yes, are these also analyzed 5.3 seconds prior to seizure onset? How was the CSD normalized? Would the oscillations of sink and source in the seizure-onset zone (SOZ) effect the normalization in the SOZ.

Figure 4 and 5 show pairing of MUA with CSD and spread pattern of seizure well.

As mentioned by the authors the pitfalls include small sampling size, seizures analyzed were predominantly temporal in onset, and they are limited by location.

Reviewer #3 (Remarks to the Author):

Title: Differential cortical layer engagement during seizure initiation and spread

Summary: The authors describe the first in vivo laminar recordings of 30 human seizures from 10 patients. I cannot overstate how invaluable this data is for understanding the neuronal underpinnings of human seizures. The manuscript is well-organized and clearly written, but with numerous small errors that need to be fixed. There are some additional details that should be added to complete the report before publication.

Minor Critiques:

Introduction:

- Line 73: 'an' should be 'can.' I would suggest rewording the first half of this sentence for clarity.
- Line 87: The scholarship in this paragraph seems sparse. More specifically, the Wenzel et al. studies are relevant here, as they characterized propagation across the lamina using two-photon neuronal imaging.
- The sentence starting on line 93 should be split in two and the second half reworded, as there is a grammatical disagreement in 'underly.'
- The introduction would be stronger if some sort of mechanistic hypothesis were stated that is derived from the literature that the authors cite.

Methods:

- Line 461: the tilde before '10-14 days' is imprecise and an actual range should be provided.
- How did the authors determine which areas were suspected of being in the ictal onset zone? (line 463)
- Grammatical errors in the sentence starting on line 470.
- What are 'stimulus markers?' (line 480)
- Line 495: There is a space missing after 2000.
- Line 496: The proportion of interpolated channels should be reported.

Results:

- It would be nice to include percentages in the quantities of discharges reported at the end of page 4 for ease of interpretation.
- There is no evidential basis for the claim in the sentence on line 144. Some quantitative analysis should be carried out to support this claim, and the seizure recorded on the laminar electrodes should be shown below that recorded on ECoG for both examples.
- Line 166: The term 'spike' is ambiguous. Replacing this word with either 'discharge' or 'action potential' when appropriate would help with clarity. This is also apparent on line 385.
- It would be nice to show a couple examples of raw MUA data.
- There's an extra comma on line 226
- There's a random equal sign on line 270

Discussion:

- Line 382: by 'empowerment' do the authors mean 'amplification?'

General:

- It would be really awesome to do a discharge-matched analysis of the patients who had laminar electrodes in and out of the seizure onset zone. That may not show much beyond what is already reported, but would be elegant.

Major Critiques:

- There are several ethical considerations involved with this work: 1) Ethical question of implanting a penetrating array into an oligodendroglioma. 2) Ethical question of implanting penetrating electrodes that are only for research. 3) Ethics of implanting an additional electrode at a distance from tissue most likely to be resected.
- The authors never showed the temporal evolution of MUA that is referred to on line 331.
- The authors should provide justification for, and more detail about, their operational definition of the ictal onset zone.
- Much more description of the ictal spike detection methods should be included. Were they detected from ECoG or from laminar probes? ECoG data is shown, but there is no mention of how ECoG data was recorded or processed in the methods.
- I'm confused by the sink/source terminology. In the ictal onset zone, the sinks reliably occur before the sources, and in the propagation zone, there are mostly sources. These results seem intuitive to me, but opposite: you would expect to see sources followed by sinks in the ictal onset zone, and only sinks in the propagation zone. Similarly, in the early stages of the seizure onset zone, I would expect there to be mostly sinks (that change to sources after recruitment), but you show mostly sources. Can you help clear up my confusion, and the potential confusion of other readers, by providing a brief explanation of this terminology.

Other questions:

- Does the sink/source pattern change before seizure termination, as ictal discharging becomes more intermittent?

REVIEWER COMMENTS AND AUTHOR'S RESPONSES

Reviewer #1 (Remarks to the Author):

Analyzing combined recordings of linear depth electrode and surface ECoG strips/grids in epilepsy patients during pre-surgical evaluation, Bourdillon and colleagues provide strong evidence for differential ictal spike dynamics across cortical layers within the presumed seizure onset zone versus propagation territories. The findings bear significant clinical value for pre-surgical diagnostic work-up in pharmacoresistent focal epilepsies, and substantiate our understanding of ictal spike generation and propagation in the human cortex. In sum, the dual laminar electrode recordings are highly appreciated, and the paper's main finding is a valuable addition to the field. There are a few things that could/should be improved.

1) General layout/narrative of the presentation of the study: when reading the manuscript in its current form, as a reader that is familiar with the matter, it felt immediately odd to look at average spike analysis across entire seizures. It's long known that seizures, and their grapho-elements, undergo an evolution throughout the seizure. Initial ictal spikes do morphologically look different to late ictal spikes. Being unsatisfied with the immediate analytical over-simplification from the get-go of the paper, I had made a bunch of comments already until I reached figure 4, where what is intuitive was now acknowledged, shown, and further analyzed. In my view, nothing speaks against having figure 4 as figure 1, as it contains everything that is shown in figure 1 and also parts of figure 2. If the paper and figure display were trimmed/re-arranged, this would provide more clarity from the beginning of the paper, and reduce redundancy of displayed CSD plots across the paper. The average analysis can still be made, as the authors find, in part, surprisingly stable CSD footprints, but the dynamic picture should be put on the table early on. Further, figure 4 has a SOZ and propagation laminar electrode from a single patient, which is the intuitive way of presentation for the reader (figure 1 shows separate patients having just one implanted laminar electrode each, which is not intuitive, if the majority of patients had two laminar electrodes, and the main interest of the paper is to contrast SOZ and propagation areas).

Response:

We thank the reviewer for their insights on our work and their constructive comments. Regarding the narrative aspect of the article, especially the presentation of the results, it has allowed us to reorganize the text. Specifically, Figure 4, which was at the end of the paper and followed the chronological order of the research process, is now at the very beginning of the results section, in the second position. This indeed provides a better hook for the reader and highlights one of the key points of the results reported here. However, we chose to keep Figure 1 in the first position, as Figure 2 is referenced almost immediately now. As the reviewer rightly pointed out, there is significant complementarity between the two figures, and with the first introducing the second, it seemed narratively challenging to reverse their positions. Naturally, the entire text and its organization regarding the results have been reviewed with this narrative perspective in mind:

The following session is now just after the “*Spike generation and spike propagation*” session:

“Dynamics in circuit activity during seizure progression

The foregoing results show the difference in involvement of the cortical layers in the generation of spikes depending on whether these are recorded directly in the ictal onset zone or in a zone of seizure propagation. However, since epileptic seizures are not static processes, we analysed the evolution of those patterns during the seizure. Inside the seizure onset zone the involvement is always granular and below with changes only in the exact pattern of sinks and sources over time. In contrast, outside the seizure onset zone the sinks and sources start above the granular layer and, while that basic motif is conserved, the initial sink of each ictal discharge involved deeper and deeper layers as the seizure progresses.

In detail, within the ictal onset zone, the CSD maxima and minima constituting the alternating sinks and sources remain in the infra-granular and granular layer throughout the seizure without any extension to the other, more superficial, layers (Fig. 2A). None-the-less, details of the pattern do change between early initiation and later phases of the seizure. Initially the sink is observed both in granular cortex / superficial infragranular cortex as well as in the deepest portion of the latter, thus flanking a source in the middle part of the infragranular cortex. After a few seconds during the seizure, the sink encompasses the entire granular cortex and the infra-granular cortex (only at the very early period of the spike generation) excluding deepest portion of the latter which constitutes the source (Fig. 1 & 2). Conversely, in the seizure propagation zones, the increase in the maximum values of the CSD seems to be confined to the supra-granular layers at the start of the seizure. This large superficial sink and slightly deeper source remain throughout the seizure in the propagation zone. However, the sink extends into deeper layers of the cortex as the seizure progresses. (Fig. 2A). It is worth saying that the sink/source pattern does not change before seizure termination, as ictal discharging becomes more intermittent.

To quantify the changing dynamics of laminar involvement more completely, we start with a detailed description of ictal discharge current flow in general comparing and contrasting events in the seizure onset zone and outside the seizure onset zone. During the generation of ictal spikes, the CSD reached its peak more frequently in supra-granular layer when recording outside the ictal onset area. Within the seizure onset area, the CSD reached its maximum in infragranular and granular layers (chi-2, $p < 0.0001$; Fig. 3A). It is worth mentioning that these observations are consistent with what has been described for inter-ictal spikes previously^{27,28} Thus, we found supra-granular, granular and infra-granular layers were not homogeneously involved in seizures. Inside the ictal onset zone, at the time of the ictal spike generation, sinks and sources were simultaneously present in granular and infra-granular layers and noticeably alternated in time: each sink at the time of the generation of a spike tended to become a source in the 50ms following the spike onset and vice versa. These oscillations between sink and source were not observed in the supra-granular layers. Similarly, in recordings made outside the ictal onset zone, such an alternation of sink and source was also observed, but only in the supra-granular layers (Fig. 3B).

To quantify these observations across all 10 patients and 30 epileptic seizures, for each patient both the variance and the normalized mean of all the positive (sinks, normalized between 0 and 1) and negative values (sources, normalized between -1 and 0) of the CSD were computed. We analysed 8 seizures in 4 patients in the ictal onset zone and the recording of 23 seizures in the propagation pathway in 8 individual patients (averaging within patients then across patients). The mean normalized CSD values were significantly higher (t-test, $p < 0.001$) within the ictal onset zone for the granular and infragranular electrodes compared to the supra-granular electrodes for both sources (-0.350 ± 0.212 vs -0.099 ± 0.098) and sinks (0.382 ± 0.249 vs 0.077 ± 0.066). Unsurprisingly this was associated with significantly greater variance (t-test, $p < 0.005$) for both sources (0.033 ± 0.035 vs 0.235 ± 0.174) and sinks (0.039 ± 0.051 vs 0.325 ± 0.202) reflecting the temporal alternation of sources and sinks in the granular and infragranular cortex within the ictal onset zone during epileptic seizures. In contrast, outside the seizure onset zone, the normalized mean CSD was significantly higher (t-test, $p < 0.001$) in the supra-granular cortex than in the other layers, both for sources (-0.507 ± 0.292 vs -0.197 ± 0.153) and sinks (0.503 ± 0.212 vs 0.112 ± 0.111). The significant difference in the analysis of variances (t-test, $p < 0.005$) between the supra-granular cortex and the other layers was in the same direction for sources (0.418 ± 0.217 vs 0.091 ± 0.097) and sinks (0.383 ± 0.221 vs 0.035 ± 0.067). The alternation of sources and sinks was therefore again found, when recording the propagation of seizures, but only in the supra-granular layers of the cortex (Figure 3B), at least at the initiation of the seizure.”

Consequently this part of the previous session has been removed:

“During the generation of ictal spikes, the CSD reached its peak more frequently in supra-granular layer when recording outside the ictal onset area. Within the seizure onset area, the CSD reached its

maximum in infragranular and granular layers (chi-2, $p < 0.0001$; Fig. 2A). It is worth mentioning that these observations are consistent with what has been described for inter-ictal spikes previously^{27,28}.

Presentation of the result of the new figure 4 is now continuing the previous paragraph:

“To examine how these patterns of laminar involvement change over time quantitatively, we utilized two different approaches. First, we performed an independent component analysis on the CSD of individual ictal discharges in all patients. Within the ictal onset zone, we identified two CSD patterns for spike generation. A first pattern was identified at the start of the seizure, on average during the first 5.3 seconds (± 1.2) whose frequency begins at 1.3 Hz (± 0.4) during the first second of the seizure and increases until 8.9 Hz (± 2.1) at the end of the period where this pattern is identified. During the rest of the seizure, a second pattern was identified lasting until the end of the seizure, 57s (± 12.4), and remained at a constant frequency of 7.1 Hz (± 0.9) except for the end of the seizure where ictal activities became intermittent. The first pattern was characterized by sinks at the deepest and most superficial level of the infragranular layers with a source in between. Immediately after ictal spike initiation, the most superficial sink and source reversed. The second pattern was characterized by a predominant sink and, during the entire pattern in the most superficial part of the infra-granular layers and a source at the deepest part. Between the two, all the remaining infragranular layers were a sink transforming very quickly into a source (Fig. 4A). In contrast, the analysis of the spikes identified in the seizure propagation zone only allowed the identification of a single pattern using the same methods (Fig. 4B).

In a second analysis, to enable a temporal comparison of the evolution of this activity during a seizure, we normalized the duration of all seizures to 100 time points. At each time point, the average absolute value (to include both sinks and sources) of the CSD was averaged for all seizures of a single patient and then across all patients. If the latter deviated by more than 2 standard deviations from the previously calculated base value, it was considered an outlier. The summation of all the values thus obtained made it possible to perform a normalized analysis of their distribution across the cortical layers through time.

Inside the ictal onset zone, no significant variation in the normalized CSD across all patients was found either within the supra-granular layers ($p = 0.391$) or within the infra-granular layers ($p = 0.052$, (Fig. 4B). Conversely, there was a significant increase ($p < 0.01$) of the normalized CSD within the granular layer between the first 24% of the seizure (mean = 0.075 \pm 0.042) and the rest of it (mean = 0.193 \pm 0.057).

Outside the ictal onset zone, in the seizure propagation areas, we noted a significant increase in the normalized CSD across all data between the 26% of the seizure and its continuation for the supra-granular layers ($p < 0.001$, mean = 0.610 \pm 0.041 vs mean 0.787 = \pm 0.069), after 28% for the granular layer, ($p < 0.001$, mean 0.008 = \pm 0.022 vs mean 0.406 = \pm 0.093) and after 42% for the infra-granular layers ($p < 0.001$, mean = 0.021 \pm 0.016 vs mean 0.057 = \pm 0.044).”

2) In the introduction, the authors give a brief, underwhelming overview of previous literature on the topic. Two of 3 referenced papers are in vitro studies, some in neonatal brain tissue, whereas other highly relevant in vitro work, and high-resolution in vivo studies on laminar ictal dynamics are not mentioned, although they exist. E.g. in vitro, see e.g. Telfeian and Connors, 1998, *Epilepsia* 39, 700–708). E.g. in vivo, there have been micoprism-assisted cellular scale in vivo imaging studies in mice on layer-specific dynamics during seizure initiation and spread. These studies are relevant to the study

presented here, the contextualization of the presented results, and the final schematic outline. Aeed et al. (Ann Neurol 2020;87:97–115) looked at seizure initiation, and found different dynamics to the ones presented here (this may however simply be due to the superficial local application of 4-AP), while Wenzel et al. (Cell Rep 2017;19:2681–2693) looked at layer-specific ictal dynamics in propagation areas in primarily unaffected cortex, and the results go in a similar direction as is reported here. Layer 4 as the entry point/leading edge during seizure propagation is also discussed there, although technical restrictions prohibited specifically addressing this possibility.

Response: We thank the reviewers for this remark and for their very helpful reference information. Our introduction was likely too concise and incorrectly omitted results from techniques other than laminar electrophysiology. To address these points, below is a new version of the paragraph:

Starting Line 82 “In vitro studies have been conducted on animal cortex in which seizures were triggered chemically by injection of antagonists of γ -aminobutyric acid A (picrotoxin). The inhibition defect caused by this injection led to a generation of horizontally propagating seizures with a crucial role of layer 5 in the propagation of seizures¹. In vivo, studies using bi-photon imaging have also been able to focus on the role of different cortical layers in the generation and propagation of seizures. In 4-aminopyridine-induced neocortical spikes and seizures mice, it was shown that seizure initiation by this method appeared to take place in the supra-granular layer². In contrast, another study in mice using two-photon calcium imaging with local field potential recordings to map cortical layers at a cell level, found different results suggesting that the spread of locally induced (4-aminopyridine or picrotoxin) relies on a supra-granular invasion followed by deep-layer recruitment during horizontal seizure spread³. Furthermore, several animal studies have been conducted using laminar electrodes and studying both local field potentials and multi-unit activity to identify the role of different cortical layers in seizure generation. In a set of animal studies using flurothyl to induce seizures, all six cortical layers were implicated in seizure generation⁴. On the other hand, in neonatal mice, seizures induced by 4-aminopyridine induce the generation of seizures in the infra-granular layers 5 and 6 while seizure propagation is observed within the supra-granular layers 2 and 3, mainly due to the activation of interneurons. Pyramidal cells were only involved in the infragranular layers during seizure generation^{5,6}. In humans, one of the very few laminar studies of epileptic activity in cortex was carried out ex-vivo and supported a role of supra-granular layers in the generation of seizure activity⁷. In contrast, laminar recordings of epileptic activity in mesial temporal structures in patients have been more consistent with excitation in the deeper pyramidal cell layer^{8–10}. From all of this work, it is difficult to build a model explaining the physiology of epilepsy, probably because chemically induced seizures within a normal animal cortex are a very different model as compare to the pathological human cortex responsible for epilepsy and which probably have a pathological functional architecture. However, it emerges from all of these results that there is indeed a different involvement of the cortical layers during seizures, thus encouraging them to be recorded during spontaneous epileptic seizures in humans.”

3) Unit activity: Could the authors distinguish between pyramidal and interneuronal units, or was this not possible, e.g. due to unit quality? This would be powerful for the contextualization of the area-specific LFP and CSD analysis performed here (e.g. fast switches between sinks and sources), and link it to cell-type specific firing dynamics. This way, one could also contextualize previous literature on cell-type specific firing dynamics during seizures within the SOZ and propagation territories, which would possibly substantiate the authors’ findings of specific LFP/CSD dynamics as a biomarker for delineating SOZ and propagation area. As unit data was obtained, it would be fantastic, if a distinction between excitatory and inhibitory units could be made.

Response: This is an excellent point. Indeed, our first idea was to use the MUA to identify the cell type involved in the activations. Unfortunately, the algorithms used for the identification of the type of neuron did not perform very well (probably due to insufficient quality and quantity of data). We agree unit type identification would have greatly helped in interpreting the data but we did not feel comfortable with those analyses without improved recording approaches. However, these results are

encouraging to the point that future data collection involving new electrodes such as Neuropixels may be necessary to answer these important questions as well as increase the volume of data and allow, we hope, to understand the activity of different types of neurons involved in the different cortical layers activities.

4) Recent studies (e.g. Smith et al., Nat Commun, 2016, doi: 10.1038/ncomms11098) suggest that the “focus”, or the “ictal core” is in fact non-stationary during seizures. So, what could be considered the “focus”, may no longer be when the seizure progresses in time and space. The results reported here give a rather “static” view of the focus. As seizures are known to be highly dynamic in space and time, this is a bit surprising. Could the authors discuss this in more detail? Could some of the changes of laminar dynamics, e.g. in propagation area, be due to a switch from “penumbra” to “ictal core”? This could, again, be substantiated e.g. by looking at pyramidal and interneuronal unit firing.

Response: Once again, we thank the reviewer for raising a very interesting point. Indeed in the excellent article from Cathy Schevon's team, data similar to those analyzed here are analyzed in Nature Communication and we originally omitted further information in the first submission on this topic. Therefore, we have added the following paragraph to the discussion comparing and contrasting the results as well as explaining these differences (below). In particular, these differences could relate to both the type of electrodes used (Utah array versus laminar electrodes) and the cortical layer access.

Line 432. “A recent study examining coupling between sub-dural recordings and 1.5 mm deep Utah Array⁴¹ microelectrode recordings show that within-ictal onset zone activity behaves like a dynamic zone. The researchers propose an interesting dynamic model in which seizures evolve into self-organized structures wherein a small seizing territory projects high-intensity electrical signals over a broad cortical areasting the importance of interactions with other distant and favourable zones in the progression of the seizure. These results may at first appear contradictory with the data reported here where we communicate a fairly static epileptic focus and spread zone during the seizures. However, further comparisons between our study and the Utah array study from smith et al. ⁴¹ demonstrate there is convergence. First, the Utah array type electrodes used in this study allow the recording of MU and LFPs in a granular layer through many recording sites. Thus the MUA is of better quality (more abundant data enabling cell type identification) but only records at one depth of the 6 cortical layers. Conversely, the electrodes used in our work have fewer recording channels (MUA more difficult to analyse) but record each of the 6 cortical layers. In the results from this study, there is a period of 5 to 10s (see Figure 5a in Smith et al. 2016)⁴¹ between pre-recruitment and the end of the ictal wavefront preceding a very homogeneous postrecruitment and pretermination periods in terms of MUA. This characteristic initial phase in MUA found in the Utah array recordings could therefore correspond to the pattern described here in the first seconds of seizure initiation (Figure 4) and precede the homogeneous activity observed throughout the seizure. The appearance of a static phenomenon in our reported data would come from the uniqueness of the laminar recording site within the epileptic focus. We hypothesize that the wavefront as reported in the Utah array recordings could have been observed if several electrodes laminar had been put within the ictal core which may be of interest in future work. However, regarding the electrodes placed in the propagation pathways, we further hypothesize the contacts were probably too distant from the ictal onset zone to be able to participate in the seizures generation and only became autonomously active very late when the deeper layers were recruited, and possibly involved in non propagated local spikes generation.

5) Methods, page 21: <<[...] When technically possible, a second laminar electrode was implanted at a distance from the first electrode. [...]>>

Please provide the concrete distances. This is important, as e.g. the propagation area may not be just one entity, but consist of distance-dependent sub-parts (e.g. penumbra or distant propagation area).

Response: Indeed, this information was lacking in the materials and methods. All of the remotely implanted electrodes were too far from the focus to be affected by the penumbra of the ictal core (continuation of the response to the previous remark). The information has been added in the Methods: *Line 529. “When technically possible, a second laminar electrode was implanted at a distance from the first electrode (always further than 5cm, see Table 1). At the end of the intracranial recording, the laminar electrode(s) were explanted during the clinical electrode removal procedure.”*

6) Results, page 4: <<[...] The CSD was maximal in the supra-granular cortex for 51,949 of these events and in the granular or infragranular cortex for the remaining 118,658 [...]>>

Is it the optimal analysis to use max values here? If one is looking for “presence” of significant changes in sources/sinks, shouldn’t this be specified with regards to interictal, or interspike interval during ictus? When using the max only, it could exclude layers that show significant changes in current flow as compared to ref. This may become especially problematic when then the spatial distribution of exclusively max CSD values is compared, as is actually done in figure 2.

Response:

Indeed, this was an interim analysis conducted at the beginning of the project, which is losing its relevance and is redundant with the information provided by figures 2 and 3 (according to the new numbering). Therefore, we have removed part A from the previous figure 2 and modified the text accordingly.

“We then identified all the events corresponding to a single epileptic spike and a normalized mean CSD across participant and cortical layers was analysed (Fig. 2 & 3). During the generation of ictal spikes, the CSD values were higher in the supra-granular layer when recording outside the ictal onset area. Within the seizure onset area, the CSD was higher in infragranular and granular layers (χ^2 , $p < 0.0001$; Fig. 2). It is worth mentioning that these observations are consistent with what has been described for inter-ictal spikes previously^{27,28}.”

7) Results, page 4: <<[...] Among the 17 laminar electrodes, 5 laminar electrode implants were considered as being in the ictal onset zone (also called the seizure onset zone), based on the clinical recording. [...]>>

Could this be contextualized in a bit more differentiated fashion, with one or two sentences? The SOZ is an inherently ill-defined zone, as one would otherwise not need intracranial electrode implantations to further delineate the factual SOZ. As a necessary consequence, and as opposed to animal research, where this can be much better controlled, one cannot be certain about whether an implanted electrode really sits where the seizure starts, or ictal spikes are generated. The placement is based on the pre-surgical diagnostic data at hand, plus an “educated guess”. The laminar electrodes have, due to their linearity, minimal x/y coverage (if the depth direction is considered as z). Thus, it’s a realistic scenario that a laminar electrode that is implanted “inside” the SOZ, is in fact not. It may simply be recruited early on, as it is close-by the SOZ. This does not make the findings shown here less relevant by any means, but it’s generally unclear whether there are just two, or more distinct ictal compartments (e.g. SOZ, nearby prop, distant prop). In how many cases were the areas, where the “SOZ laminar electrodes” were implanted, factually surgically removed subsequently, with good outcome? This would be a posthoc prove that the electrodes had indeed been situated in the SOZ, and it would be a relevant information to put into this paper, as it would indeed strengthen the usefulness of the findings in a biomarker context, especially for patients with pharmacoresistent “non-lesional” focal epilepsies.

Response: We thank the reviewers for their discussion and they rightly point out the ongoing difficulty of defining SOZ as well as the lack of certainty surrounding electrode locations, particularly the laminar array. To address these points and improve clarity, we expanded on our description of the techniques.

If the laminar electrode was indeed implanted on the basis of a hypothesis of the SOZ location, the classification as ictal onset zone or not was based on the analysis of nearby conventional intracranial recordings. In other words, we used the recording with the implanted clinical subdural grid to confirm that the laminar electrode was indeed in the zone most precociously affected by the seizure. Furthermore, the analysis of the electrophysiological clinical data did not suggest that the ictal onset zone could have been outside the region explored by the subdural grid. We added in material and methods:

Line 533... “At the end of the intracranial recording, all the clinical intracranial electrophysiological data were analysed by the epileptology and clinical team to identify the ictal zone. First of all, the neurophysiologist made sure that nothing could suggest that the ictal onset zone is located outside the field of exploration of the subdural grid. If this was the case, the epileptogenic zone was then identified and it was possible to determine whether the laminar electrodes were located inside or outside the ictal onset zone.

8) Results, page 8: <<[...] We analysed 8 seizures in 4 patients in the ictal onset zone and the recording of 23 seizures in the propagation pathway of 8 individual patients [...]>>

Does that mean that in 3 patients, no seizure was recorded in the SOZ although they had two laminar electrodes? Does that mean that the intra-SOZ laminar electrodes were actually not in the SOZ in 3 patients? How were these electrodes analytically dealt with (spatial re-assignment)? This also refers back to the general issue of the ill-defined SOZ.I

Response: Indeed, two (and not 3, see table 1) patients had two laminar electrodes implanted each, both being outside the SOZ. During the implantation of each of these patients (Patient 3 and Patient 7), the surgeon implanted one of the electrodes in an area considered to be the ictal onset zone based on the preoperative hypothesis. However, the analysis of the intracranial recording carried out for clinical purposes showed that the hypothesis was not supported and that the laminar electrode was neither in the SOZ nor even in the penumbra and, in the analyses, we assigned these electrodes as outside the SOZ or ictal zone. The answer to this remark, as the reviewer pointed out, was provided in the text by the previous answer.

9) At times, the readers have to reconstruct some of the numbers/aspects themselves, which disturbs the reading process unnecessarily. Simply write it out, please, e.g.:

Results, page 4: <<[...] Seven out of the 10 patient had two laminar electrodes implanted. [...]>>

The three remaining patients could have each 1 or >2 electrodes. Only the next sentence indirectly reveals that it must be a single electrode each for the remaining 3 patients. Suggestion: Seven out of the 10 patient had two laminar electrodes implanted, three patients a single laminar electrode.

Or results, page 4: <<[...] The CSD was maximal in the supra-granular cortex for 51,949 of these events and in the granular or infragranular cortex for the remaining 118,658 [...]>>

Please make it easier for the reader by e.g. adding “Inside the presumed onset zone,” at the beginning of the sentence. The sentence before does not talk about inside/outside, and the sentence before that one mentions both, so the reader has to add up the numbers themselves (in looking at the following sentence where “Outside the ictal onset” is specified), to find out that the authors are talking about SOZ here.

Response: We would like to thank the reviewer for their very helpful input and suggested corrections to improve the manuscript. Changes have been made in the text taking into account the suggestions made as below:

Line 133. “Seven out of the 10 patients had two laminar electrodes implanted with three patients a single laminar electrode each.”

Line 152.. “Inside the presumed onset zone , the CSD was maximal in the supra-granular cortex for 51,949 of these events and in the granular or infragranular cortex for the remaining 118,658 (see Methods for determination of cortical layer and electrode relationship).”

10) Grammar/Typos/misspelled words; e.g.:

- Cover page: National Center for Neurological Disorders, , Xuanwu Hospital, Clinical
- Page 1: These latter electrodes an have multiple form factors
- Results, page 4: Seven out of the 10 patient had two laminar electrodes
- Methods: evaluation were offered the participate
- Table 1: Hippocampic
- Methods: LFP recordings was thus recorded on 3.5 mm
- Methods: The LFP and MUA were recorded simultaneously at 2000and 20000 Hz.
- Figure 1, legend: <> Maybe I am mistaken here, but shouldn't it read “granular” cortex here?
- Figure 2, legend: <<[...] for electrodes implanted in the ictal onset zone (left) and in an area of spread (right). [...]>> Left and right and top and bottom are not correctly labeled here I assume.

Response: We thank the reviewer again for their vigilance and for taking the time to make these suggestions. We have made these corrections throughout the text (see revised manuscript).

11) Figure legend, figure 1: Please specify red arrow; this is misleading with the red shading of the SOZ just above, in the figure. Of course, the arrow is temporal, not spatial, but still, some readers may be confused briefly.

Response: Indeed, this point has been corrected.

12) “Spike generation” is numerous times across the paper, at times in a way that may trigger confusion or misunderstandings in the reader, please re-word in unambiguous/clear terms. Some examples:

- Results, page 8: <<[...] To better understand how spike generation differs within and outside ictal onset zones at the cortical level [...]>>

“Spike generation” is intuitively tied to a point of generation from where it then spreads. The wording here suggests an innumerable number of generators located inside and outside the SOZ. Possible solution: “How spike dynamics differ within” and so on.

- Results, page 4: <<[...] During the generation of ictal spikes, the CSD reached its peak more frequently in supra-granular layer when recording outside the ictal onset area. [...]>> Each spike is inherently generated. Here, the authors are not analyzing spike generation per se, but simply the CSD during, or across individual spikes in the presumed SOZ and outside of it.

Response: We thank the reviewers for their valuable input and suggestions. We had originally used this term as addressed in a previous paper (Fabo *Brain* 2008), but we understand the confusion this could represent. It has become evident since that publication that it can now cause confusion. Following the

reviewer's recommendation, we have replaced it with 'discharge' (modification of all occurrence in the text and in the figures) which is more accurate and recommended as well by reviewer 2.

Reviewer #2 (Remarks to the Author):

This is an excellent and essential paper to understand seizure generation in human cortex. I have a few comments.

1) Fig 2; does the "ictal spikes" refer to pre-seizure epileptic sharps. If yes, are these also analyzed 5.3 seconds prior to seizure onset?

Response: We thank the reviewer for their positive comments as well as for their critical questions. Concerning the 'ictal spikes': these are only the spikes occurring during the seizure itself and not the pre-seizure epileptic sharps. We thank the reviewer for pointing out this possible ambiguity. The legend of the figure has been altered to reduce the ambiguity.

Line 181.. "Figure 2 : Distribution of CSD values across cortical layers. A: Distribution of the maximal CSD value during ictal spikes across the 24 contacts of the laminar electrodes (0 is the pial surface) during the seizure. Events recorded outside the ictal onset zone are plotted in orange, and those recorded inside in blue. N= 25,568 events. B: Top: Normalized mean across subjects and seizures of the CSD for sources (in blue) and sinks (in red) for electrodes implanted in the ictal onset zone (left) and in an area of spread (right). The CSD is not limited to the ictal spikes but is computed on the whole signal during the seizures. Samples of CSD raw data with the same colour scale as for figure 3 and recorded in supra and infra-granular are represented to illustrate the temporal alternation of sources and sinks. Bottom: Example from single participants of the sink/source alternation over 1s during a seizure."

2) How was the CSD normalized? Would the oscillations of sink and source in the seizure-onset zone (SOZ) effect the normalization in the SOZ.

Response: If the reviewer refers to the normalization made for the comparison of the maximum and minimum CSDs (continuation of the previous comment), this was done separately for the negative and positive values so that the sink and source influence did not have any effect on the normalization. The approach was used to ensure that the values were comparable between the participants. The text has been adapted.

Line 583: The equation for rescaling data X to an arbitrary interval [0 1] was:

$$X_{rescaled} = 1 + \left[\frac{X - \min_x}{\max_x - \min_x} \right] (0 - 1)$$

3) Figure 4 and 5 show pairing of MUA with CSD and spread pattern of seizure well. As mentioned by the authors the pitfalls include small sampling size, seizures analyzed were predominantly temporal in onset, and they are limited by location.

Response: This is indeed a pitfall which we address in the discussion. Unfortunately, this issue is difficult to overcome due to the low amount of human data, which are difficult to acquire. We hope this limitation is sufficiently well-addressed.

Reviewer #3 (Remarks to the Author):

Title: Differential cortical layer engagement during seizure initiation and spread

Summary: The authors describe the first in vivo laminar recordings of 30 human seizures from 10 patients. I cannot overstate how invaluable this data is for understanding the neuronal underpinnings of human seizures. The manuscript is well-organized and clearly written, but with numerous small errors that need to be fixed. There are some additional details that should be added to complete the report before publication.

Minor Critiques:

Response: We thank the reviewer for these general glowing comments and for their constructive review of the article. Regarding the minor critiques, here (in italic) are the changes to improve the manuscript.

Introduction:

- Line 73: 'an' should be 'can.' I would suggest rewording the first half of this sentence for clarity.

Line 72. "Among these electrodes two main types are developed: on the one surface electrodes allowing neural recording parallel to the surface of the cortex with a spatial resolution of less than 50 μm and on the other hand electrodes penetrating the cortical surface to record perpendicular to the cortical surface across the different layers of the grey matter^{14,15}"

- Line 87: The scholarship in this paragraph seems sparse. More specifically, the Wenzel et al. studies are relevant here, as they characterized propagation across the lamina using two-photon neuronal imaging.

We thank the reviewer and have now addressed this point in a related comment by Reviewer 1. We have made modifications in response, particularly with the addition of a paragraph starting on line 82 including much more background information.

- The sentence starting on line 93 should be split in two and the second half reworded, as there is a grammatical disagreement in 'underly.'

Line 114... "These laminar recordings allowed estimation of the synaptic activity (represented by local field potentials, LFPs) and somatic action potentials (multi-unit activity) within each of the six cortical layers. This makes it possible to describe the microcircuitry of the neocortex that underly the generation and propagation of seizures."

- The introduction would be stronger if some sort of mechanistic hypothesis were stated that is derived from the literature that the authors cite.

We again thank the reviewer and have now addressed this point in a related comment by Reviewer 1. Much of this information is also included in paragraph starting on line 82 including.

Methods:

- Line 461: the tilde before '10-14 days' is imprecise and an actual range should be provided.

We apologize for the lack of precision. The number of days implanted was actually the range for all 10 patients. Modifications have been made as such: Line 522. "The implantation of the electrodes for clinical purposes was not conditioned or influenced by the study. Patients were implanted with subdural grid electrodes (Ad-Tech Medical Instrument corporation, Racine, WI, USA) and the recordings continuously obtained (range from 10 to 14 days in the 10 participants) at a sampling rate of 256 or 512Hz"

- How did the authors determine which areas were suspected of being in the ictal onset zone? (line 463)

Line 525... "The implantation of the laminar electrodes was done in areas suspected of being the ictal onset zone (cortex likely to be resected) based on phase I investigation including clinical examination of seizures, long lasting video scalp EEG, high resolution MRI, magnetoencephalography and metabolic imaging."

- Grammatical errors in the sentence starting on line 470.

We thank the reviewers for their comments. They are in line with additional suggestions made by Reviewer 1 and we have therefore edited as appropriate.

- What are 'stimulus markers?' (line 480)

Line 533.. "Stimulus markers (i.e. electrical pulse generated artificially and sent simultaneously to the two EEG amplifiers) were used to synchronize the recordings from the multielectrode array and clinical macroelectrodes."

- Line 495: There is a space missing after 2000.

We thank the reviewers for their comments. They are in line with additional suggestions made by Reviewer 1 and we have therefore edited as appropriate.

- Line 496: The proportion of interpolated channels should be reported.

We have addressed this with the below edits: Line 569... "The LFP and MUA were recorded simultaneously at 2000 and 20000 Hz. A filtering was then applied online from 0.2 to 500 Hz and 200 to 5,000 Hz. Data from channel considered rejected after visual inspection of data were linearly interpolated from the channels directly above and below them (12.5% of channels)."

Results:

- It would be nice to include percentages in the quantities of discharges reported at the end of page 4 for ease of interpretation.

We thank the reviewer for their input. We have addressed this with the below edits: Line 146... "A total of 614,071 ictal spikes were identified across all electrode channels (408 channels) while 170,607 (27.8%) spikes were identified in the seizure onset zone. Current Source Density (CSD) measurement has been used to localize the source of the LFPs within the cortical layers and primarily reflects post-synaptic activity (see Methods)15.... Inside the presumed onset zone, the CSD was maximal in the supra-granular cortex for 51,949 (30.4%) of these events and in the granular or infragranular cortex for the remaining 118,658 (69.6%) (see Methods for determination of cortical layer and electrode relationship). Outside the ictal onset zone, we identified 443,494 events, 391,398 (88.2%) having a maximum CSD in upper supra-granular cortex and 52,096 (11.8%) in the deeper granular or infragranular cortex."

- There is no evidential basis for the claim in the sentence on line 144. Some quantitative analysis should be carried out to support this claim, and the seizure recorded on the laminar electrodes should be shown below that recorded on ECoG for both examples.

We thank the reviewers for their comments. They are in line with additional suggestions made by Reviewer 1 and we have therefore edited as appropriate

The ECoG and the Laminar recording are shown on figure 1, but it was unclear that the red arrow was pointing the seizure onset on the same time scale for the two modalities of intracranial recordings.

We apologize for the confusion. We have edited the caption as listed below: Line 169. "Seizure onset (red arrow) on laminar recordings is similar to the adjacent ECoG electrodes. Ictal activities were first observed in granular and infra-granular layer recordings (sources are plotted in blue) with sinks both in the adjacent granular cortex and in the deepest infragranular cortex. Later during the seizure, the source moved into the deepest layer and the sink is constituted by all the remaining granular and infragranular cortex. B: The laminar electrode (yellow dot) was implanted outside the ictal onset zone. Seizure onset (red arrow) on laminar recordings is late as compared to the adjacent ECoG electrodes."

- Line 166: The term 'spike' is ambiguous. Replacing this word with either 'discharge' or 'action potential' when appropriate would help with clarity. This is also apparent on line 385.

This comment aligns with the first reviewer's observation, and the change has already been made.

- It would be nice to show a couple examples of raw MUA data.

Thank you for the suggestion. A sample of raw MUA data has been provided.

- There's an extra comma on line 226

This has been corrected.

- There's a random equal sign on line 270

This has been corrected.

Discussion:

- Line 382: by 'empowerment' do the authors mean 'amplification?'

Line 417.. "Deep layer current flow later in the seizure, in particular granular and superficial infragranular, could correspond to an autonomization within the seizure propagation zone of the generation of epileptic activities."

General:

- It would be really awesome to do a discharge-matched analysis of the patients who had laminar electrodes in and out of the seizure onset zone. That may not show much beyond what is already reported, but would be elegant.

Thank you very much for this comment. If part B of figure 4 indeed shows an analysis at the scale of all the participants, the part A of the figure shows an analysis at the level of the single participant with a discharge-matched analysis for the two electrodes in and out of the seizure onset zone. In order to remove any ambiguity, we have re-specified this in the legend of figure 4.

Line 293. "Figure 4: Activity evolves through time across layers during a seizure. A: Example in a single participant having two laminar electrodes of change of the CSD over time for a seizure recorded within the ictal onset zone (top) and outside the ictal onset zone (bottom). The start and end of seizures are marked with red arrows. The red lines represent the CSD distribution across the cortex and the clear change in the CSD spread for seizures outside the onset zone. Insets in the middle show an expanded view of the CSD at different time points. B: Average representation of CSD evolution across all participants and seizures during seizures recorded within the ictal onset zone (upper matrix) and outside of it (bottom matrix). The values correspond to the normalized distribution of the absolute values of CSD (thus combining sink and source) differing significantly from the baseline."

Major Critiques:

- There are several ethical considerations involved with this work: 1) Ethical question of implanting a penetrating array into an oligodendroglioma. 2) Ethical question of implanting penetrating electrodes that are only for research. 3) Ethics of implanting an additional electrode at a distance from tissue most likely to be resected.

All protocols that have been approved by the local Institutional Review Boards (IRBs) in accordance with ethical practices of human subjects research. Electrode implantation was carried out in areas **assumed to be removed during surgery**, sometimes being distant from the ictal onset zone. It is worth noting that this research, utilizing innovative intracranial electrodes, falls within the framework of the recently addressed and specific reflections on intracranial electrophysiological research ('Ethical commitments, principles, and practices guiding intracranial neuroscientific research in humans' by Ashley Feinsinger et al. and 'Which Ethical Issues Need to Be Considered Related to Microwires or Utah Arrays?' by Young). Furthermore, our team recently published another study on human laminar electrophysiology within the Nature group ('Large-scale neural recordings with single neuron resolution using Neuropixels probes in human cortex' by Paulk et al.), which proved to be fully compliant with the IRB and the ethical requirements of the Nature group journals.

- The authors never showed the temporal evolution of MUA that is referred to on line 331.

To couple this suggestion with the previous one concerning the MUA and not to multiply the figures, the temporal evolution of the MUA has been shown through the raw MUA signal presented in the new figure previously mentioned.

- The authors should provide justification for, and more detail about, their operational definition of the ictal onset zone.

We thank the reviewer for their comment. We have attempted to clarify in the text. If the electrode was indeed implanted on the basis of a hypothesis, the classification as ictal onset zone or not was based on the analysis of conventional intracranial recordings. In other words, the recording with the clinical subdural-grid put in place during the implantation should have

shown that the laminar electrode was indeed in the zone most precociously affected by the seizure. Furthermore there should be no elements that could suggest that the subdural grid used for the detection of the ictal onset zone was not placed in an optimal way (ictal onset zone located outside the field of exploration). We added in material and methods:

Line 533... “At the end of the intracranial recording, all the clinical intracranial electrophysiological data were analysed by the epileptology team. First of all, the neurophysiologist made sure that nothing could suggest that the ictal onset zone is located outside the field of exploration of the subdural grid. If this was the case, the epileptogenic zone was then identified and it was possible to determine whether the laminar electrodes were located inside or outside the ictal onset zone.

- Much more description of the ictal spike detection methods should be included. Were they detected from ECoG or from laminar probes? ECoG data is shown, but there is no mention of how ECoG data was recorded or processed in the methods.

We thank the reviewer for their comment. We attempted to expand upon our description of the method. The ictal spike detection was carried out by adapting published scripts (Chu 2017 and Kramer and Naladin 2021) and are available on Github (<https://github.com/Mark-Kramer/Spike-Ripple-Detector-Method/commits?author=Mark-Kramer>). The detection was conducted directly on the laminar recordings. We have included this information in the manuscript.

Regarding ECoG with clinical subdural grid, recording was performed with a clinical Compumedics or Micromed System at a sampling frequency of 512Hz. The description has been improved in the Methods section:

Line 579... “An automated ictal spike detection was used to locate the spikes and measure the CSD at the time of the spikes as well as the average of the CSD over a period of 100ms centered by the spike. The automatic spike detection was based on a previously published method (<https://github.com/Mark-Kramer/Spike-Ripple-Detector-Method/commits?author=Mark-Kramer>). The spike detection was performed directly on the laminar recordings. Potential gradient modification was thresholded, and the speed of its modification (ascending and descending) was taken into account. A minimum event duration was also an adjustable parameter so as not to wrongly include artefacts. A visual control was carried out on a sample of 100 spikes for each patient to obtain a sensitivity and a specificity greater than 90%.”

- I'm confused by the sink/source terminology. In the ictal onset zone, the sinks reliably occur before the sources, and in the propagation zone, there are mostly sources. These results seem intuitive to me, but opposite: you would expect to see sources followed by sinks in the ictal onset zone, and only sinks in the propagation zone. Similarly, in the early stages of the seizure onset zone, I would expect there to be mostly sinks (that change to sources after recruitment), but you show mostly sources. Can you help clear up my confusion, and the potential confusion of other readers, by providing a brief explanation of this terminology.

We thank the reviewer for highlighting the confusion regarding the terminology, particularly when applied to laminar recordings. When we speak of sink and source, it refers to the secondary spatial derivative of field potentials, the method used to estimate the CSD. In other words, at a time t , there are sinks and sources at the same time and this represents the current flows between the cortical layers, or the layers involved at the time of a spike. Temporal alternation can certainly occur between spikes for a given area. We have clarified this in the manuscript:

Line 150... "The calculation of the CSD produce at each time t , sinks and sources making it possible to locate the current flow between the cortical layers and therefore those which are involved at that time."

Other questions:

- Does the sink/source pattern change before seizure termination, as ictal discharging becomes more intermittent?

This is an excellent point and we had not explicitly specified this point. Surprisingly, and contrary to the first seconds of the seizure, we did not note any modification of the sink/source patterns at the end of the seizure, just before the seizure. This is now specified in the manuscript as below:

Line 290.. "It is worth saying that the sink/source pattern does not change before seizure termination, as ictal discharging becomes more intermittent"

REVIEWER COMMENTS

Reviewer #1 (Remarks to the Author):

The effort of the authors to address the reviewers' comments is much appreciated. The revised manuscript has clearly improved in structure, presentation of the results, narrative, and contextualization with previous literature. In light of studies on the dynamic nature of seizures, ictal cores, or the transition from interictal to ictal conditions (e.g. Huberfeld and Menendez de la Prida et al, Nat Neuro, 2011), this reviewer still has difficulties to wrap their head around some of the described stationary features of ictal discharges, or ictal discharges vs. previously reported interictal discharges (e.g. lines 223-224). Yet, this only reflects that there is much more to be learned about human seizure generation and spread. The paper clearly presents a highly valuable and important contribution to the field of epilepsy research.

Remaining minor comment:

Oddly, the manuscript seems to have more errors regarding Grammar/Typos/misspelled words than the previous version. Please go through the manuscript again carefully, and correct where necessary. Just a few examples:

- line 88: discharges and seizures IN mice
 - line 119: onset seizure zone -> seizure onset zone
 - line 134: Seven out of the 10 patients had two laminar electrodes implanted with three patients a single laminar electrode each -> syntax/punctuation
 - line 141: [...] clinical team reviewed by an independent expert. Missing comma
 - line 152: [...] Methods)15.Briefly [...] missing space
 - line 154: <> fragment of a sentence
- And so on...

Reviewer #2 (Remarks to the Author):

The authors have answered all the concerns appropriately. I have nothing further to add.

Reviewer #3 (Remarks to the Author):

The manuscript is improved from the last time I read it. However, there are some new errors, and one critique that remains insufficiently addressed.

Major critiques:

- Figure 4 is cut off, so it is hard to critically assess, and may not fit on a page. In the 'ICA pattern detection' section of figure 4, there appear to be several rasters that extend over the x-axis. Why does that occur?

Minor critiques:

- Line 83: make the 'y' in GABA a gamma. It would be nice to use a Greek letter, rather than the English shorthand in several other places throughout the manuscript as well.
- Line 85: the citation #19 is too big.
- Line 93: The word 'underly' is an archaic form of the word 'underlie.' You may want to change it on that line and on line 118, where there is another apparent disagreement in number.
- Lines 152-154 have several punctuation errors.
- There is another typo in the title of figure 5B.

REVIEWER COMMENTS AND AUTHOR'S RESPONSES

Reviewer #1 (Remarks to the Author):

Oddly, the manuscript seems to have more errors regarding Grammar/Typos/misspelled words than the previous version. Please go through the manuscript again carefully, and correct where necessary. Just a few examples:

-line 88: discharges and seizures IN mice

-line 119: onset seizure zone -> seizure onset zone

-line 134: Seven out of the 10 patients had two laminar electrodes implanted with three patients a single laminar electrode each -> syntax/punctuation

-line 141: [...] clinical team reviewed by an independent expert. Missing comma

-line 152: [...] Methods)15. Briefly [...] missing space

-line 154: <> fragment of a sentence

And so on...

Response:

We thank the reviewer for highlighting the grammatical and typographical errors. These have been corrected, and we have ensured that a thorough proofreading was conducted by multiple authors before submission.

Reviewer #2 (Remarks to the Author):

The authors have answered all the concerns appropriately. I have nothing further to add.

Response:

We thank the reviewer for taking the time to review the article and for enabling its improvement through the comments from the first round of review.

Reviewer #3 (Remarks to the Author):

Major critiques:

- Figure 4 is cut off, so it is hard to critically assess, and may not fit on a page. In the 'ICA pattern detection' section of figure 4, there appear to be several rasters that extend over the x-axis. Why does that occur?

We apologize for the formatting error in Figure 4, which appears to have occurred during the final conversion to PDF. This has been corrected. The interruption on the x-axis has also been fixed and was related to a vector layer from a previous version. This has been rectified. The code (Jupyter) used to generate the figure (ICA and plot) has been included in the revision (ICAfigure.ipynb), along with the data used for this figure (csd_cleandata_seizure_1.mat).

Minor critiques:

- Line 83: make the 'y' in GABA a gamma. It would be nice to use a Greek letter, rather than the English shorthand in several other places throughout the manuscript as well.
- Line 85: the citation #19 is too big.
- Line 93: The word 'underly' is an archaic form of the word 'underlie.' You may want to change it on that line and on line 118, where there is another apparent disagreement in number.
- Lines 152-154 have several punctuation errors.
- There is another typo in the title of figure 5B.

We thank the reviewer for highlighting the grammatical and typographical errors. As mentioned in the response to reviewer #1, who also found multiple such errors, we have had multiple authors thoroughly proofread the final version and hope these and other issues have been resolved.

REVIEWERS' COMMENTS

Reviewer #3 (Remarks to the Author):

the authors have reasonably assuaged my concerns

REVIEWER COMMENTS AND AUTHOR'S RESPONSES

Reviewer #3 (Remarks to the Author):

Reviewer #3 (Remarks to the Author):

The authors have reasonably assuaged my concerns

Response:

We thank the reviewer for taking the time to review the article and for enabling its improvement through the comments from the first round of review.